# Maternal high-fat diet in mice induces cerebrovascular, microglial and long-term behavioural alterations in offspring

Maude Bordeleau[1,2], Cesar H. Comin[3], Lourdes Fernández de Cossío[1,4], Chloé Lacabanne[1], Moises Freitas-Andrade[5,6,7], Fernando González Ibáñez[2,8,9], Joanna Raman-Nair[5,6,7], Michael Wakem[10], Mallar Chakravarty[1,11,12,13], Luciano da F. Costa[14], Baptiste Lacoste [5,6,7,17✉] & Marie-Ève Tremblay [2,8,9,15,16,17✉]

Various environmental exposures during pregnancy, like maternal diet, can compromise, at critical periods of development, the neurovascular maturation of the offspring. Foetal exposure to maternal high-fat diet (mHFD), common to Western societies, has been shown to disturb neurovascular development in neonates and long-term permeability of the neurovasculature. Nevertheless, the effects of mHFD on the offspring's cerebrovascular health remains largely elusive. Here, we sought to address this knowledge gap by using a translational mouse model of mHFD exposure. Three-dimensional and ultrastructure analysis of the neurovascular unit (vasculature and parenchymal cells) in mHFD-exposed offspring revealed major alterations of the neurovascular organization and metabolism. These alterations were accompanied by changes in the expression of genes involved in metabolism and immunity, indicating that neurovascular changes may result from abnormal brain metabolism and immune regulation. In addition, mHFD-exposed offspring showed persisting behavioural alterations reminiscent of neurodevelopmental disorders, specifically an increase in stereotyped and repetitive behaviours into adulthood.

[1] Integrated Program in Neuroscience, McGill University, Montreal, QC, Canada. [2] Neurosciences Axis, CRCHU de Québec-Université Laval, Québec, QC, Canada. [3] Department of Computer Science, Federal University of São Carlos, São Carlos, SP, Brazil. [4] Department of Neurosciences, University of California, La Jolla, San Diego, CA, USA. [5] Department of Cellular and Molecular Medicine, Faculty of Medicine, University of Ottawa, Ottawa, ON, Canada. [6] University of Ottawa Brain and Mind Research Institute, Ottawa, ON, Canada. [7] Ottawa Hospital Research Institute, Neuroscience Program, Ottawa, ON, Canada. [8] Département de médecine moléculaire, Université Laval, Québec, QC, Canada. [9] Division of Medical Sciences, University of Victoria, Victoria, BC, Canada. [10] Genetic Sciences Division, Thermo Fisher Scientific, Burlington, ON, Canada. [11] Cerebral Imaging Center, Douglas Mental Health University, McGill University, Montréal, QC, Canada. [12] Department of Psychiatry, McGill University, Montréal, QC, Canada. [13] Department of Biological and Biomedical Engineering, McGill University, Montréal, QC, Canada. [14] São Carlos Institute of Physics, University of São Paulo, São Carlos, SP, Brazil. [15] Department of Neurology and Neurosurgery, McGill University, Montréal, QC, Canada. [16] Department of Biochemistry and Molecular Biology, The University of British Columbia, Vancouver, BC, Canada. [17] These authors contributed equally: Baptiste Lacoste, Marie-Ève Tremblay. ✉email: blacoste@uottawa.ca; evetremblay@uvic.ca

Pre- and early postnatal developmental periods are times of significant organisation and reorganisation of the multi-level architecture of the central nervous system (CNS). Specifically, the CNS vasculature provides the infrastructure for the energy exchange that the brain requires to efficiently function. Its development is initiated during embryogenesis, with the invasion of the perineural plexus into the neuroectoderm[1] and expands postnatally into refined and mature vascular networks comprising arteries, veins and capillaries[1]. The cerebrovasculature interacts closely with parenchymal cells thereby forming a complex multicellular system known as the neurovascular unit (NVU), in which endothelial cells and pericytes, but also parenchymal neurons, astrocytes and microglia cooperate to regulate brain development, function and homoeostasis[2,3]. This process is key as the brain is the most resource-demanding organ, accounting for up to a quarter of the body's total energy consumption[4]. However, various environmental risk factors during critical neurovascular developmental stages, like maternal diet, can compromise neurovascular maturation[5], conferring risk for a range of neurodevelopmental and neuropsychiatric disorders[6].

Foetal exposure to maternal high-fat diet (mHFD) is an increasing problem, as ~40 million pregnant women worldwide are considered overweight, in part due to an excess dietary intake of fat (a known risk factor for pregnancy complications and developmental alterations)[7]. In both the mother and offspring, maternal high-fat diet (mHFD) has been shown to promote inflammation[8], and, during pregnancy, this represents a key risk factor for an array of neurodevelopmental and neuropsychiatric disorders[8,9]. However, the effects of mHFD on the offspring's cerebrovascular health are still understudied. So far, it has been demonstrated that maternal obesity disturbs development of the blood-brain barrier in neonate rodent offspring[10] as well as leads to an increased vascular permeability in the hypothalamus of foetal, neonatal[10] and adult[11,12] rodent offspring. Even though neurovascular alterations are hypothesised as a main mechanism linking maternal immune activation to the onset of neurodevelopmental and neuropsychiatric disorders[9], the effects of mHFD on the neurovascular system of brain regions involved in these disorders have remained elusive.

Here, we provide a comprehensive assessment of mHFD consequences on the NVU organisation in a translational mouse model of mHFD exposure (Supplementary Fig. 1). Using an approach that combines state-of-the-art imaging and transcriptomics-based techniques, we evaluated the impact of mHFD within the mouse cerebral cortex and hippocampus during adolescence, which coincides with a critical stage of immune maturation and neurovascular network refinement[9,13]. Our work notably identified mHFD-driven cortical hypervascularization associated with an increase of perivascular microglia, as well as important immune and metabolic changes at the ultrastructure and transcriptomic levels. Together, these changes may have contributed in part to the development of stereotypic and repetitive behaviours assessed by marble burying at adulthood, suggesting that mHFD-mediated neurovascular alterations might be partly involved in the expression of behaviours reminiscent of neurodevelopmental disorders[14].

## Results and discussion
Using our validated method combining microscopy and computational image analysis[3,13], we first sought to characterise the 3D structure of blood vessels at adolescence, in terms of its organisation and its complexity (Supplementary Fig. 1b). We quantified vascular density (mm/mm$^3$), number of branch points (per mm$^3$), as well as vessel tortuosity (i.e., index of voxel-distance deviating to the medial axis) (Fig. 1a–h, j–m, Supplementary data 1). Vascular density and the number of branch points were significantly increased in mHFD-exposed offspring compared to standard chow diet (CD)-exposed offspring (Fig. 1, Supplementary data 1). Tortuosity index of cortical blood vessels, however, remained unchanged following mHFD exposure (Supplementary data 1). In the hippocampus, blood vessel density, branching and tortuosity were also similar between offspring groups (Fig. 1i, n, Supplementary data 1), indicating a region-specific effect of mHFD exposure. To evaluate whether this change in cortical cerebrovascular system organisation was associated with functional alterations of blood vessels, we characterised the ultrastructural properties of capillaries in the parietal cortex by quantifying the area of the endothelium, basement membrane thickness, number of mitochondria, fusion/fission rate of mitochondria, as well as number, length, and thickness of tight junctions (Supplementary Fig. 1c). Surprisingly, these parameters remained unchanged in the mHFD-exposed offspring (Supplementary Table 1), suggesting that hypervascularization of the cerebrovasculature was not associated with major dysfunction of cortical endothelial cells.

Changes in blood vessel structure can result from direct alteration of endothelial cells or dysfunction of pericytes[14], cells from a mesodermal origin that line up the CNS endothelium[2]. While endothelial cells create the vascular wall[2,15,16], pericytes are mural cells required for formation and maintenance of brain blood vessels during embryonic development, formation and integrity of blood-brain barrier, as well as the regulation of cerebral blood flow during postnatal life[2]. Quantification of PDGFRβ-positive pericyte coverage revealed that the observed differences in cortical vessels were not associated with a change of pericyte coverage at adolescence (Supplementary Fig. 2a–n). This was further confirmed by assessment of pericyte coverage and anchoring using nanoscale-resolution electron microscopy (Supplementary Fig. 2o–r, Supplementary Table 1). Nevertheless, besides pericytes, other cellular elements of the NVU could be impacting the development of the endothelium, like glial cells.

Microglia, the resident immune cells of the brain from a mesodermal origin, are emerging elements of the NVU[2] that contribute to the *glia limitans* surrounding arterial, venous and capillary vessels[17,18]. They are notably implicated in blood vessel branch fusion and regression, as well as endothelial sprouting during neurodevelopment, and blood flow regulation throughout life[1,19]. We evaluated the impact of mHFD on Iba1-positive (+) microglial density, average distance to blood vessels, and ratio of cells contacting a vessel during adolescence. In all regions investigated, microglial density remained unaffected (Supplementary data 1). However, in the cortex, microglia from mHFD-exposed offspring were closer to blood vessels ($P = 0.003$; Fig. 2a–h) and made more putative contacts with blood vessels ($P = 0.005$; Fig. 2a–d, j–m) compared to CD-exposed offspring. The anterior and parietal, but not occipital, subdivisions of the cortex, also displayed similar differences (Fig. 2f–h, k–m, Supplementary data 1). These changes in microglia-vascular interactions were associated with a significant decrease in microglial spacing index in the anterior cortex of mHFD- compared to CD-exposed offspring ($P = 0.034$; Supplementary Fig. 3a–d, t), indicating an uneven microglial distribution that could leave parenchymal areas unattended. In the hippocampus, microglia from control CD-exposed offspring were farther from blood vessels in male compared to female offspring ($P = 0.018$; Fig. 2i, Supplementary Fig. 4). Similarly, the ratio of microglia contacting blood vessels was increased in CD-exposed female *versus* male offspring ($P = 0.027$; Fig. 2n, Supplementary Fig. 4), indicating that microglia-blood vessel interactions differ between sexes in the normal adolescent offspring and in a region-specific manner. Of note, cortical microglial proximity and putative contacts with blood vessels were significantly different between female and male

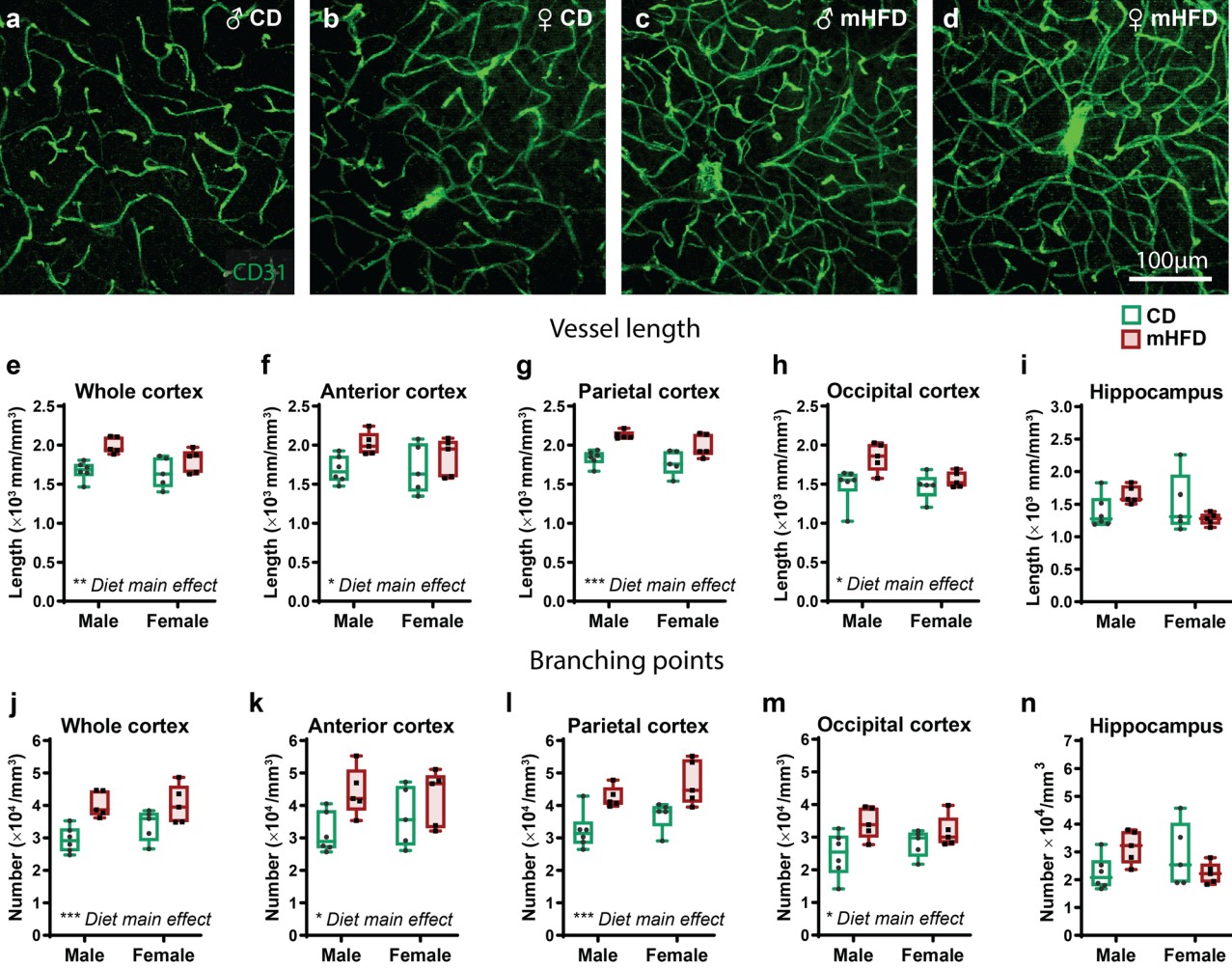

**Fig. 1 mHFD promotes increased length and number of branching of cortical blood vessels in P30 offspring. a–d** Zoom-in of pictures taken at 20x with an epifluorescence microscope illustrate blood vessels organisation and complexity in the parietal cortex of adolescent offspring. **e–i** Vessel length and (**j–n**) branching points of blood vessels in the (**e, j**) whole cortex, (**f, k**) anterior cortex, (**g, l**) parietal cortex, (**h, m**) occipital cortex, and (**i, n**) hippocampus of the adolescent offspring were increased after exposure to mHFD. Whisker graphs show minimum, median (line) and maximum, as well as individual data points ($n$ = 5–6 animals/diet/sex). *$P < 0.05$, **$P < 0.01$, ***$P < 0.001$ by 2-way ANOVA analysis (all data, except occipital cortex vessel length), *$P < 0.05$ by mixed-effects analysis (for occipital cortex vessel length). ♀: female, ♂: male, CD: control diet, mHFD: maternal high-fat diet.

offspring (Proximity: whole cortex: $P = 0.003$; anterior: $P = 0.006$; parietal: $P = 0.003$; occipital: $P = 0.009$; Putative contacts: whole cortex: $P = 0.001$; anterior: $P = 0.007$; parietal: $P = 0.009$; occipital: $P = 0.003$; Fig. 2a–m), indicating global sex differences between cortical microglia. Microglia from the parietal cortex also showed an even distribution in male compared to female offspring, regardless of maternal diet (Male vs Female; $F_{(1,17)} = 4.948$, $P = 0.039$; Supplementary Fig. 3e–h, u). Further characterisation of microglia-blood vessel interactions within the parietal cortex by electron microscopy revealed at nanoscale resolution that these proximities did not involve direct contacts between microglial cell bodies and the basement membrane of capillaries, with less contacts actually observed upon mHFD (Fig. 2o–s, Supplementary Table 1). Considering previous findings that microglia closely associate with endothelial tip cells during angiogenesis and vessel fusion[20], our observations suggest that increased microglial proximity to blood vessels might take part in the cortical hypervascularization of mHFD-exposed adolescent offspring. Similarly, microglial recruitment to vessels was reported in different contexts of inflammation across animal models and in humans[17]. Thus, this increase in microglial proximity to cortical vessels could result from exacerbated

inflammation in the offspring brain, as previously observed upon exposure to mHFD[8,21]. Under these inflammatory conditions, microglia might then release several immune-related mediators, including chemokines and cytokines, which can in turn influence the neurovascular organisation[19,22,23].

To provide further insights into microglial function, we evaluated ultrastructural changes of their organelles and direct interactions with their microenvironment in the parietal cortex. Upon mHFD, cortical microglia had an increased number of mitochondria specifically in males, while showing a loss of sex differences in cellular stress markers (dilation of endoplasmic reticulum and Golgi apparatus cisternae) and number of lipid bodies contrary to the control offspring (Fig. 3a–g, Supplementary Table 2). More specifically, cellular stress markers were more frequently observed in mHFD-exposed female offspring reaching a level similar to the male offspring (Fig. 3a–d, f, Supplementary Table 2). The number of lipid bodies in mHFD-exposed male offspring was also reduced to reach the number observed in female littermates (Fig. 3a–d, g, Supplementary Table 2). Furthermore, while looking at microglial interactions with their microenvironment, we found that microglia from mHFD-exposed male offspring made more contacts with presynaptic

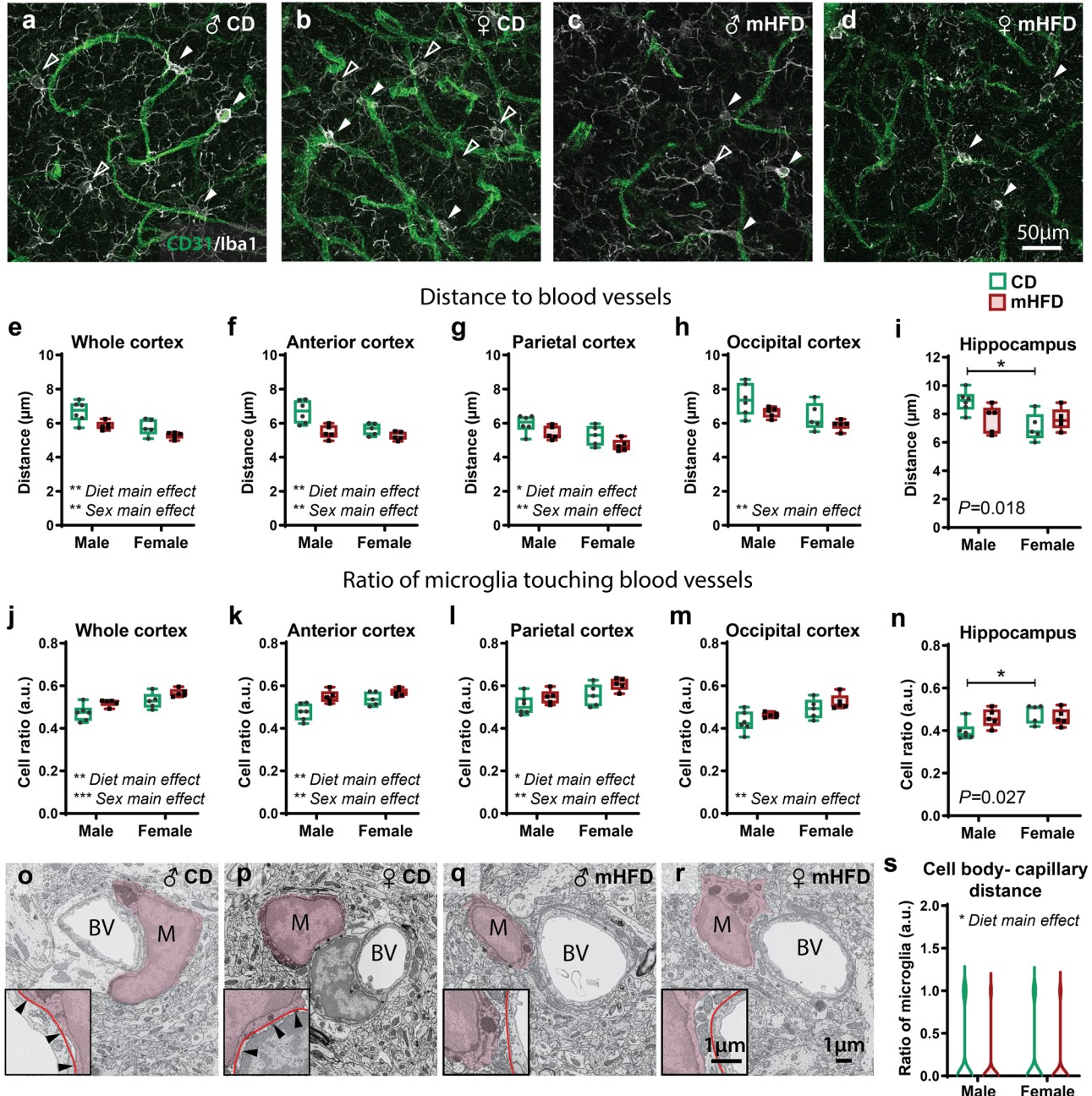

**Fig. 2 Microglial distance to blood vessels and their putative contact with blood vessels in P30 offspring are impacted in a region-specific manner.**
**a–d** Pictures taken at 63x with a confocal microscope show microglial interaction with blood vessels from the parietal cortex, where microglia making putative contacts (filled white arrowheads) and microglia that are not making putative contacts (open white arrowheads) were identified. **e–i** Distance to blood vessels and (**j–n**) ratio of microglia touching blood vessels were calculated for the (**e, j**) whole cortex, (**f, k**) anterior cortex, (**g, l**) parietal cortex, (**h, m**) occipital cortex and (**l, n**) hippocampus of the adolescent offspring ($n = 5$–6 animals/diet/sex). **o–r** Micrographs show average distance of microglial cell body (identified by "M", pseudocoloured in fuchsia) from capillaries (identified "BV") of the parietal cortex, where direct contact with the basement membrane (line in red) are indicated by a filled black arrowhead. (**s**) Analysis at the nanoscale level revealed reduction of these direct contacts by microglial cell body ($n = 77$–84 microglia/diet/sex, $N = 4$ animals/group). Whisker graphs show minimum, median (line) and maximum, as well as individual data points ($n = 5$–6 animals/diet/sex). Violin graphs show minimum, median (line) and maximum (77–84 microglia/diet/sex, $N = 4$ animals/group for ultrastructure analysis of microglia). *$P < 0.05$, **$P < 0.01$, ***$P < 0.001$ by 2-way ANOVA analysis (for cortical regions). *$P < 0.05$ by 2-way ANOVA analysis followed by Bonferroni post-hoc (for the hippocampus). *$P < 0.05$ by mixed-effects analysis (for ultrastructure analysis). ♀: female, ♂: male, a.u.: arbitrary units, CD: control diet, mHFD: maternal high-fat diet.

axon terminals, which led to sex differences in microglial interactions with both presynaptic and postsynaptic elements when comparing male and female offspring exposed to mHFD (Fig. 3a–d, h–I, Supplementary Table 2). Other than changes in microglia-synapse interactions, a reduced number of microglia-

associated extracellular space pockets, considered essential to microglial motility[24,25] and known to fluctuate during experience-dependent neuronal remodelling[26,27], was observed after mHFD exposure in male offspring suggesting a shift in their surveillance and/or synaptic remodelling capacity. By influencing

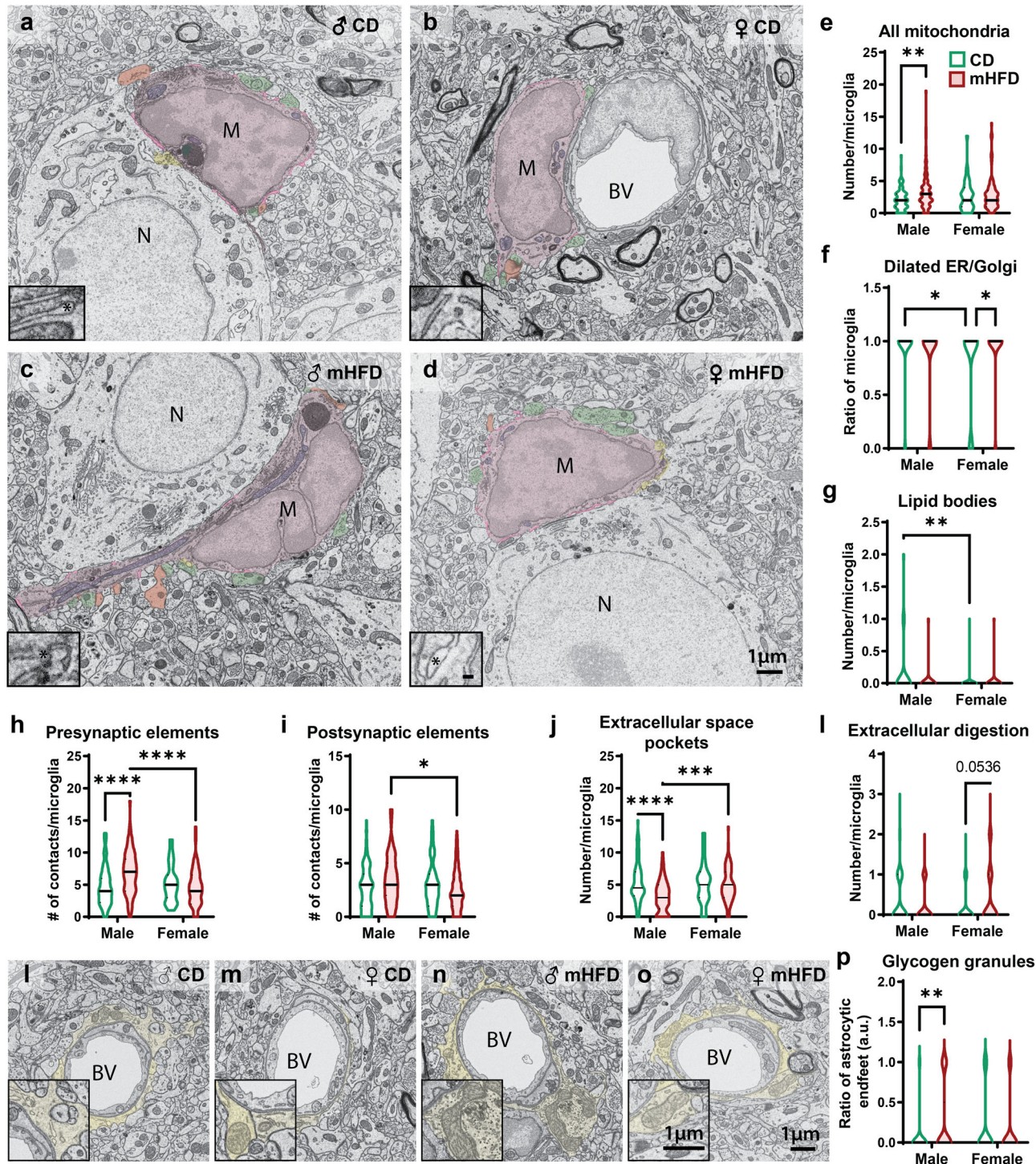

**Fig. 3 Microglia and astrocytic endfeet are sexually dimorphically affected in P30 offspring exposed to mHFD. a–d** Pictures illustrate ultrastrucutral changes of microglia (identified by "M", pseudocoloured in fuchsia), where the insets show a higher magnification on endoplasmic reticulum and/or Golgi apparatus cisternae in which the scale bar is equivalent to 100 nm (visible in panel **d**). Changes were notably observed in (**e**) number of mitochondria (pseudocoulored in purple), (**f**) dilation of the endoplasmic reticulum and Golgi apparatus cisternae (identified by "*"), and (**g**) lipid bodies (pseudocoloured in turquoise) of microglial cell bodies as well as their interactions with (**h**) presynaptic axon terminals (pseudocoloured in green), (**i**) postsynaptic dendritic spines (pseudocoloured in red), (**j**) extracellular space pockets (pseudocoloured in pink) and (**k**) extracellular digestion also known as exophagy (pseudocoloured in dark yellow). **l–o** Micrographs show representative astrocytic endfeet (pseudocoloured in yellow) for each offspring group, where mHFD-exposed male offspring have an (**p**) increased accumulation of glycogen granules. Violin graphs show minimum, median (line) and maximum (77–84 microglia/diet/sex, N = 4 animals/group for ultrastructure analysis of microglia; n = 93–95 capillaries/diet/sex for astrocytic glycogen granules analysis). *P < 0.05, **P < 0.01 by mixed-effect analysis followed by Bonferroni *post-hoc*. ♀: female, ♂: male, a.u.: arbitrary unit, CD: control diet, mHFD: maternal high-fat diet.

synaptic function, microglia may also indirectly participate in modulating angiogenesis of the brain through a phenomenon called "neurovascular coupling", where neuronal activity leads to an increase in cerebral blood flow[28–31]. There was also a trend for microglia from female offspring exposed to mHFD to increase their digestive exophagy, a lysosomal exocytosis process that promotes digestion of large cellular and subcellular elements by microglia[32,33] (Fig. 3l, Supplementary Table 2). However, this trend did not reach significance and replication of this finding would be needed in future studies. Together, our findings suggest that cortical microglia of male offspring show a shift in their metabolism, motility as well as interactions with presynaptic axon terminals, while microglia from female offspring are less globally impacted but show modifications of cellular stress, thus highlighting sex-specific changes of microglia. Indeed, previous work by our team and others have identified sex differences in the consequences of mHFD on the offspring throughout their life, and among other brain regions[11,21,34–36]. Of note, regardless of the maternal diet, microglia presented sex differences in organelles involved in metabolism (i.e., mitochondrial alterations) and phagocytosis (i.e., tertiary lysosomes, empty endosomes) (Supplementary Fig. 5, Supplementary Table 2), further highlighting the sexual dimorphism of microglia and the importance of considering both sexes throughout developmental stages[21,37–43].

Nevertheless, it remains possible that brain cells other than microglia also play a role in mHFD-induced NVU organisation changes, including astrocytes. Ultrastructural characterisation of astrocytic main parameters (coverage, average size and glycogen granules accumulation in endfeet) in the parietal cortex identified an increased accumulation of glycogen granules in mHFD-exposed male offspring (Fig. 3l–p, Supplementary Table 1).

To identify underlying pathways involved in mHFD-induced NVU changes in the cortex (i.e., hypervascularization, increased microglia-blood vessel proximity), we further assessed gene expression by whole-transcriptome microarray technology Clariom™ S using cortical extracts from adolescent offspring. Considering that no sex difference was identified in the hypervascularization of the cerebrocortex, whole transcript expression was analysed regardless of the offspring sex. Transcriptome analysis identified significantly decreased levels of 11 transcripts and significantly increased levels of four transcripts (Fig. 4a). Out of these 15 transcripts, four belong to the haemoglobin pathway. *Hba-a2;Hba-a1*, *Hbb-bs;Hbb-b1*, *Hbb-bt;Hbb-b2* were reduced by 3.19 to 5.58-fold, and *Erdr1* increased by 16.48-fold, in the mHFD-exposed offspring (Fig. 4b, c). In endothelial cells, these factors are implicated in the regulation of nitric oxide signalling[44] which regulates angiogenesis and vascular reactivity[45]. In the brain, haemoglobins were mainly found to be expressed by parenchymal neurons[46], where they were demonstrated to support mitochondrial function[47]. At lower levels, haemoglobins are also expressed by microglia[46], however, their function in these cells remains to be defined. Nevertheless, our results point toward metabolic and functional changes in cortical cells including microglia, which showed a change in organelles associated with both metabolism and cellular stress in mHFD-exposed offspring of both sexes. In fact, similar important increase of *Erdr1* was also previously reported in the CNS (retina) of mice lacking microglial fractalkine receptor, CX3CR1, notably involved in microglia-mediated synaptic pruning[48]. Furthermore, dysregulation of both haemoglobin genes and *Erdr1* was previously reported in neurodevelopmental disorders, such as autism spectrum disorders (ASD) and attention-deficit-hyperactivity-disorder[49,50]. In addition, *Cyp7a1*, involved in cytochrome P450 and lipid metabolism[51], was found to be downregulated in offspring exposed to mHFD (Fig. 4b, c). In addition to its metabolic function among several brain cell types, reduced expression of

cytochrome P450 component has been demonstrated to result in enhanced inflammatory responses[52]. Thus, besides a shift of microglial metabolism, the increased number of mitochondria found in cortical microglia may participate in a compensatory mechanism to limit the reduced expression of mitochondrial genes like cytochrome P450. Other than transcripts from metabolic pathways, tumour suppressor genes (*Btg2*)[53] and genes involved in immunity (*Csprs*, *Igtp*)[54,55] were changed (Fig. 4b, c), highlighting potential cell cycle- and immune-regulation alterations in the cortex of offspring exposed to mHFD. Out of all the transcriptomic changes, *Igtp* expression was the most robustly increased after exposure to mHFD, as indicated by its lower *P* value. Moreover, its expression was confirmed to be expressed by a majority of Iba1+ cortical microglia (CD: 80.5% of microglia vs mHFD: 83.8% of microglia), as revealed by fluorescence in situ hybridization combined with immunofluorescence labelling (FISH-immunofluorescence; Fig. 4d). Together with our ultrastructural findings, these results highlight an immune dysregulation, in which microglia may actively participate *via* functional alterations of their physiological roles in helping neurodevelopmental processes like angiogenesis and/or synaptic pruning[8,19,21]. However, it remains difficult to discriminate or conclude how these different pathways and genes are involved in underlying the observed vascular changes, which would warrant future studies to specifically look at these different targets.

To determine long-term consequences of these changes on behaviour, we performed a battery of behavioural tests, revealing that motor function, spatial working memory, social preference, social novelty preference, as well as anxiety-like and sensorimotor gating, were unchanged in adult offspring exposed to mHFD (Supplementary Fig. 6, Supplementary Table 3). Nevertheless, when compared to their CD-exposed controls, mHFD-exposed offspring of both sexes buried almost twice as many marbles during the marble burying test which is usually used to asses anxiety-related behaviour as well as repetitive and stereotypic behaviour (P = 0.004; Fig. 5, Supplementary Table 3), a common feature of several neurodevelopmental and neuropsychiatric disorders[56].

Previously, only a few studies have investigated the effects of mHFD on the brain vasculature and NVU. In foetal and neonatal mouse offspring, Kim et al. reported an increased blood vessel permeability associated with increased expression of tight junction genes (i.e., Claudin1, Claudin3, Zona Occludens 1) and endothelial transporters genes (i.e., low-density lipoprotein-related receptor protein 1, dysferlin) as well as enlarged vascular fenestrations in the hypothalamic arcuate nucleus[10]. Similarly, Contu et al. reported in aged mice that blood vessels display increased coverage by Iba1+ microglia expressing CD68, a marker of phagolysosomal activity, in the hippocampus of male offspring exposed to HFD throughout life[11]. This remodelling was associated with increased blood vessel leakiness, measured by IgG extravasation, in the hippocampus of aged female offspring exposed to HFD throughout life[11]. Another investigation focused on remodelling of the middle cerebral artery showed that mHFD causes arterial wall thickening together with increased blood pressure in adult rat offspring[12]. Here, we reveal that mHFD leads to increased blood vessel density and branching (i.e., hypervascularization) accompanied by increased microglial proximity to vessels across the cerebral cortex but not the hippocampus, of adolescent offspring in both sexes.

In fact, the microenvironmental cues regulating endothelial cell migration and proliferation are necessary for proper functional angiogenesis in vitro[16]. When migration speed was disproportionate to proliferation speed, this resulted in reduced vascular function[16]. In the context of mHFD-induced hypervascularization, our data support the hypothesis that functional alterations of immune cells, particularly the resident microglia, are an active

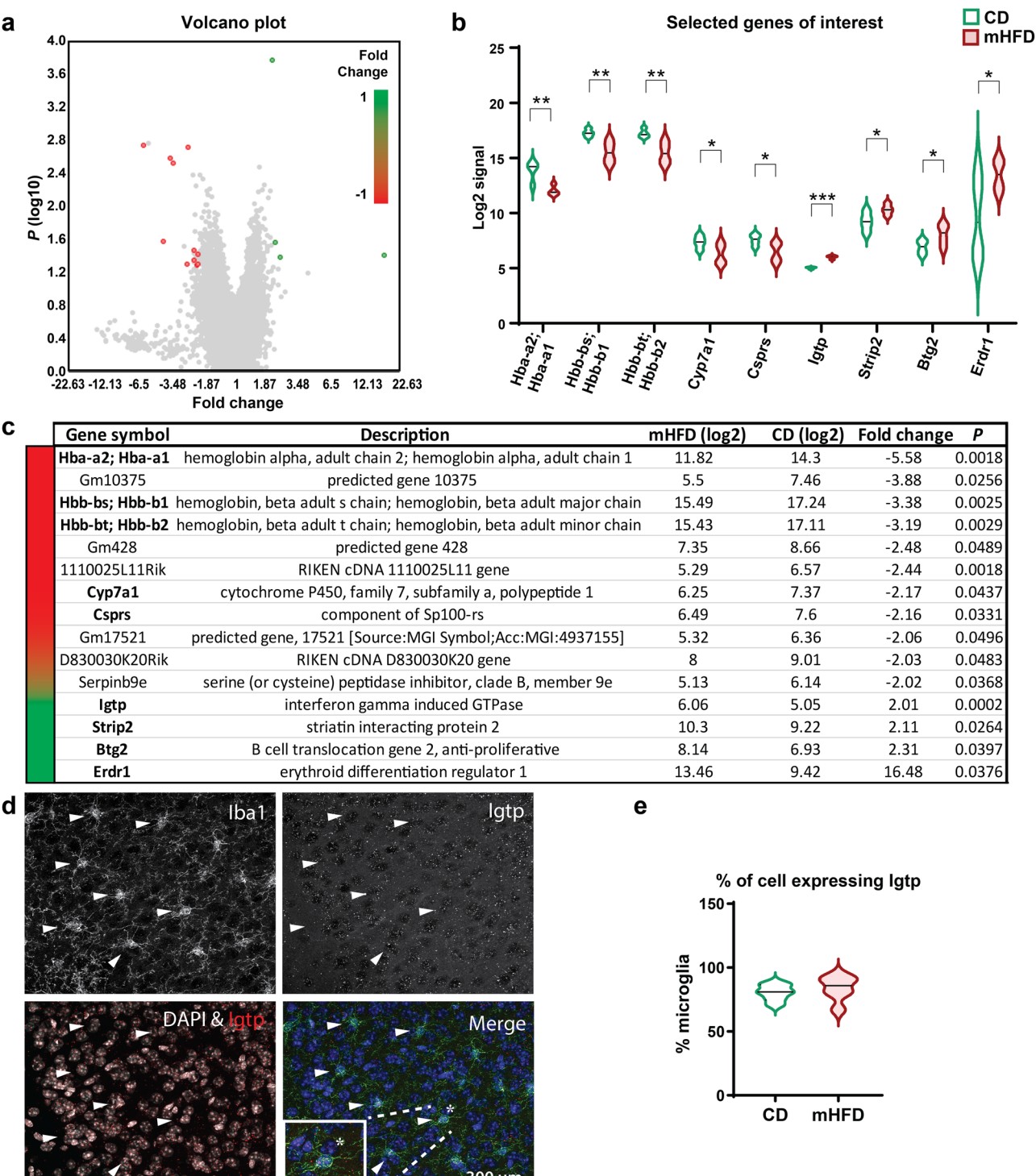

**Fig. 4 RNA microarray reveals differential expression of 15 genes in the cortex of P30 offspring exposed to mHFD, regardless of their sex. a** Volcano plot of *P* value versus fold change help visualise 15 genes with differential expression; four upregulated and eleven downregulated on whole-genome after mHFD. **b** Selected genes of interest include genes of the haemoglobin pathway (*Hba-a2;Hba-a1, Hbb-bs;Hbb-b1, Hbb-bt;Hbb-b2, Erdr1*), cytochrome P450 pathway (*Cyp7a1*), immunity (*Csprs, Igtp*), cytoskeleton (*Strip2*) and tumour suppressors (*Csprs, Btg2*). **c** Exhaustive table of differentially expressed genes with relative expression, fold change and *P* value are detailed in the bottom part (*n* = 4 animals/diet). **d** FISH-immunofluorescence allowed us to visualise *Igtp* (in red), the most affected gene, expressed in Iba1[+] microglia (in green) within their nucleus (DAPI in blue). *Igtp*-positive Iba1[+] cells are indicated by white arrowheads. **e** Percentage of microglia expressing *Igtp* per animal was calculated (*n* = 6 animals/diet). Males and females are grouped together since no sex differences were found for the cerebral cortical vasculature. Violin graphs show minimum, median (black line) and maximum. *\*P* < 0.05, *\*\*P* < 0.01, *\*\*\*P* < 0.001 by empirical Bayes statistical analysis for differential expression. CD: control diet, FISH: fluorescence in situ hybridization, mHFD: maternal high-fat diet.

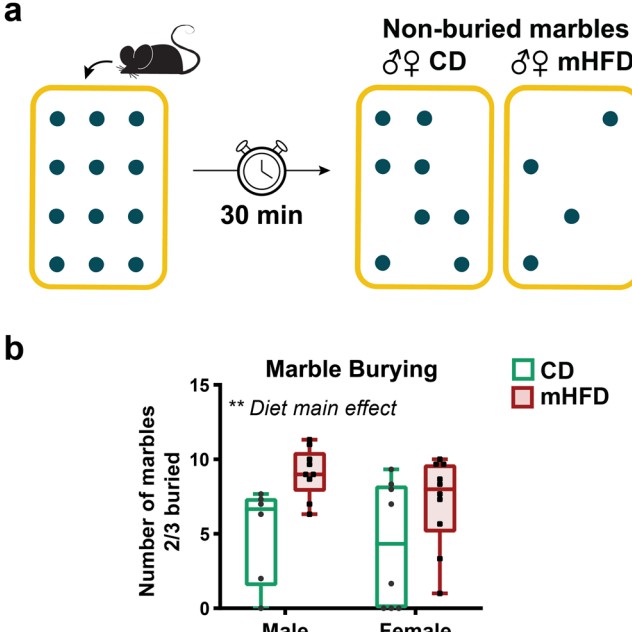

**Fig. 5 mHFD promotes an increase of stereotypic and repetitive behaviours in the marble burying task of adult offspring. a** Marbles burying paradigm is illustrated in the upper panel. **b** Results of the task are presented in the lower panel. Whisker graph shows minimum, median (line) and maximum, as well as individual data points ($n = 6$–10 litters/diet/sex, 8–12 animals/group). **$P < 0.01$ by 2-way ANOVA. ♀: female, ♂: male, CD: control diet, mHFD: maternal high-fat diet.

player in driving cortical hypervascularization. Moreover, microglia increase their interactions with the vasculature during proangiogenic processes[20], which might be partially driven by mHFD through metabolic and inflammatory changes within the cortex. Although whole-cortex transcriptomics prevented us from identifying cell-specific alterations, our findings taken with the observation of altered cerebrovascular structure and microglial ultrastructural changes suggest that metabolic and immune regulation alterations may account, at least in part, and in a sex-specific manner, for mHFD-induced pathological changes.

The results obtained in the marble burying test together with the absence of anxiety-like behaviours point towards increased stereotypic and repetitive behaviours. Such stereotypic behaviours are common in patients with ASD and obsessive-compulsive disorders (OCD)[56], among other neurodevelopmental and neuropsychiatric disorders of genetic and environmental origin, with an early life onset[14]. In the current study, the neurovascular alterations and increased microglial proximity that we observed took place within cortico-limbic regions, known to be associated with stereotyped and repetitive behaviours, and where hypervascularization was previously reported *post-mortem* in ASD brain samples[14]. While Ouellette et al. recently reported reduced cerebral angiogenesis in a genetic ASD mouse model[3], the role of neurovascular changes in other genetic or environmental ASD models, in genetic or pharmacologically induced OCD models, and in OCD patients' *post-mortem* samples remains to be determined[14]. Nevertheless, neurovascular alterations, observed as a reduction or increase of cerebrovasculature, alongside changes in the NVU, may impact neurovascular coupling, and thus contribute to changes in glial activities and functions[2,13], including the augmentation of microglial-synapses interaction reported here.

Overall, our work reveals the profound impact that excess fat, through mHFD exposure, exerts on the development of the

offspring's neurovascular system, leading to changes in vascular structure and microglial-vessel coverage, metabolism- and immune-regulating gene expression. While we observed stereotypic and repetitive behaviours in adulthood, most behavioural parameters remained normal suggesting the encouraging possibility that mHFD-induced vascular and microglial alterations may be part of a beneficial adaptation that results in a relatively minimal impact on the long-term phenotype. Similar mHFD-driven cerebrovascular modifications of the NVU in humans could render, at least in part, the offspring prone to developing a wide variety of neurodevelopmental and neuropsychiatric disorders, which warrants further investigation of the combined impact of environmental and genetic factors. Hence, appraising the influence of mHFD on the cerebrovasculature stresses the importance of implementing precautionary measures to notably reduce the "priming" effect that mHFD may have on the progeny, while studying the mechanisms involved and identifying therapeutic targets for children and adolescents at risk.

## Methods

Animal protocol was approved by McGill University's Facility Animal Care Committee under the guidelines of the Canadian Council on Animal Care. Animals were submitted to classic housing condition: 12 h dark/light cycle (8:00–20:00) with free access to water and food. C57BL/6 N female mice aged 5 weeks were obtained from Charles River and habituated to pair-housing for one week. A total of seven staggered cohorts of female mice were used for litter breeding ($n = 86$–114 females/diet). At 6 weeks, female mice were either fed with control chow (CD; Teklad 2014, detailed in Supplementary Table 4; ENVIGO, Indianapolis, IN, United States) or a HFD (diet rich in saturated/unsaturated fat; 60% kcal by lipids, detailed in Supplementary Table 4; Teklad TD.06414, ENVIGO) *ad libitum* for 4 weeks to mating, then throughout gestation and nurturing, up to weaning of their litter. Throughout the diet protocol, dams consumed equivalent caloric and protein intake, while the fat intake was increased to the detriment of carbohydrates, as previously reported[21].

After mating, pregnant dams were housed single or alone with their litter. CD and HFD dams had a rate of pregnancy of 38.4% and 51.8%, respectively, and both groups of dams showed similar pregnancy duration and litter size (number of total offspring, male and female pups). Of note, dams fed with HFD dams tended to cannibalise more often their litter without reaching statistical significance (37.3% vs 21.2%). At postnatal day (P)21, offspring were weaned, split by sex two to five per cage and put on CD. Weaned offspring had similar body weight and showed a similarly low prevalence of developmental anomalies, as previously observed[21]. Animal with major developmental anomalies (unopened or anormal eyes, dwarfs or severe dental malformation) were excluded from subsequent protocol (Fig. 1a). For each experiment, 1–2 offspring were used per litter to prevent litter-driven effect, and animals were randomly assigned for histology, transcriptomic, ultrastructure or behavioural experiments.

A first cohort of P30 offspring were used for tissue collection for histology analysis ($n = 5$–6 animals/diet/sex). Animals were sacrificed by cervical dislocation without anaesthesia, decapitated and their brains were extracted. Brains were split into hemispheres from which the mesencephalon was removed. These hemispheres containing the cortices and hippocampus were flattened between two slides, then post-fixed in 4% paraformaldehyde (PFA) diluted in phosphate buffer (PB, ~16 h; pH = 7.4) overnight (~16 h) at 4 °C. Flatten cortices were rinsed three times for 5 min in phosphate-buffered saline (PBS, 50 mM), agarose-embedded, and cut in 120 μm sections using a VT1200S vibratome (Leica Biosystem, Wetzlar, Germany) and stored in cryoprotectant solution (30% ethylene glycol, 30% glycerol in PBS) at −20 °C until use.

For transcriptomics, offspring were anaesthetised with a rodent cocktail (0.3 mL/100 g) containing ketamine [100 mg/mL], xylazine [20 mg/mL] and aceprozamine [10 mg/mL], then either fresh-decapitated for RNA processing ($n = 4$ animals regardless of sex/diet) or perfused with ~15 mL of ice-cold PBS and with ~150 mL of 4% PFA for FISH-immunofluorescence ($n = 5$–6 animals/diet/sex). For RNA processing, the animals were fresh-decapitated, rinsed with ice-cold RNAse-free PBS, their brain extracted, and their cortex dissected, flash-frozen on dry ice and stored at −80 °C until processing for whole transcript expression analysis by Clariom™ S. For FISH-immunofluorescence, brains were extracted, post-fixed for 24 h, switched to 30% sucrose solution (in PBS) for 72 h, frozen, and cut (thickness 30 μm) using a CM3050S cryostat (Leica Biosystem, Wetzlar, Germany). Cryosections were collected and stored in RNAse-free cryoprotectant solution at −20 °C until use.

For ultrastructure analyses, offspring ($n = 4$ animals/diet/sex) were anaesthetised with rodent cocktail before perfusion with ~15 mL of ice-cold PBS, ~75 mL of 3.5% acrolein in PB and ~125 mL of 4% PFA. Brains were extracted, post-fixed for 2 h and cut into 50 μm thick section using a VT1200S vibratome (Leica). Brain sections were then collected and stored in cryoprotectant solution at −20 °C until use.

One last cohort of offspring (composed of four staggered breeding cohorts, $n = 6–10$ litters/diet/sex, $N = 8–12$ animals) was used to evaluate functional behavioural outcomes in adulthood (P60-P85) and determine the long-term consequences of mHFD exposure on the offspring.

**Histology.** Flattened-brain sections were rinsed three times in PB to wash out the cryoprotectant solution from the tissues, then blocked for 2 h at room temperature (RT) in blocking solution containing 10% donkey serum and 0.5% fish-gelatine in 0.5% PB-triton (PBT). Sections were then incubated overnight at 4 °C with a cocktail of primary antibodies: rat α-CD31 (1:200, cat# 550274, BD Pharmigen, San Diego, CA, United States), guinea pig α-vGLUT2 for layer IV barrels (1:300, cat# AB2551-I, MilliporeSigma, Burlington, MA, United States), goat α-PDGFβ receptor (1:300, cat# AF1042, R&D Systems, Minneapolis, MN, United States), rabbit α-Iba1 (1:1000, cat# 019-19741, FUJIFILM Wako Chemicals, Osaka, Japan) diluted in blocking buffer. The next day, sections were rinsed three times in 0.5% PBT and incubated at RT for 3 h in a cocktail of secondary antibodies diluted in blocking solution donkey α-rat A488-conjugated (1:300, cat# A21208, Life Technologies, Carlsbad, CA, United States), donkey α-guinea pig AMCA-conjugated (1:300, cat# 706-156-148, Jackson ImmunoResearch, West Grove, PA), donkey α-goat A568-conjugated (1:300, cat# A11057, Life Technologies), donkey α-rabbit A647-conjugated (1:300, cat# A31573, Life Technologies). Immunostained sections were then rinsed twice in 0.5% PBT and twice in PB before mounting on slide. Mounted sections were let dry for an hour, quick-washed in milli-Q water and coverslipped in fluorescence protecting mounting medium FluoromountG. Slides were dried for 1 h at RT and stored at 4 °C until imaging.

Three sections of the flattened cortices (anterior, parietal, and occipital cortices) were imaged respectively corresponding to cortical layers II/II, IV and V, while two sections of the flattenhippocampus CA1 were imaged. Images were acquired with Zen 2 Pro software in z-stacks (1 picture/μm for ~54 μm) at ×10 by optical sectioning using a structured illumination ApoTome 2 mounted on an Axio Imager M2 epifluorescence microscope equipped with an AxioCam 506 mono (Zeiss, Oberkochen, Germany). Z-stacks were merged into 2D plan for spacing and nearest neighbour distance (NDD) microglial cells analysis, while unmerged z-stacks were used for 3D analysis of blood vessel endothelium, pericytes and microglial analyses (i.e., density ratio, distance to blood vessels and ratio contacting blood vessels). For microglial proximity and putative contacts, representative images were taken at 40x with a AxioImager Z2 confocal microscope equipped with a LSM800 camera (Zeiss).

For 2D microglia analysis, microglial density, NND and spacing index (density × NND²) were measured using a semi-automated manner using ImageJ software (v1.51j8; National Institute of Health, Bethesda, MD, United States). For each picture, a mask of Iba1⁺ cells was created and processed with 'Analyse particles' plugin to calculate cell count. Then, 'NND' plugin was run to determine NND value for each cell and average per picture. Spacing index was calculated with automatic cell density count and NND values, as previously described[21].

For 3D endothelium, pericyte and microglia analysis, a computational analysis of the NVU, including endothelial cells, pericytes and microglia, was performed as previously described[3,13]. $\bar{I}_v r = 40\,\mu m v\,\bar{I}_v$ were classified as belonging to a blood vessel. Connected components with a volume smaller than 500 μm³ were removed from the image. The medial axes of the segmented blood vessels were then obtained using the Palágyi-Kuba thinning algorithm[57]. Voxels in the medial axis having three or more 8-connected neighbours were classified as branching points, while voxels having only one neighbour were considered termination points (Supplementary Fig. 1b). The total length of the blood vessels in an image was calculated as the sum of the arc-lengths of the medial axes divided by the image volume. The number of branching points was also normalised by image volume. For tortuosity quantification, each voxel in the medial axis was associated with a segment, composed of all voxels having a Euclidean distance smaller than 20 μm from the reference voxel. Then, a straight line was fitted to the voxels in the segment using least squares regression, and the average voxel to line distance for the voxels in the segment was used for defining the tortuosity of the blood vessel at each point in the medial axes. The overall tortuosity of the blood vessels in an image was obtained as the average tortuosity calculated for the voxels.

Regarding pericyte quantification, the segmentation step was the same as in the case of blood vessels. Using the same parameters for the segmentation has the advantage of making the resulting blood vessels and pericytes binary images more compatible for the analysis (Supplementary Fig. 1b). For quantifying the percentage of blood vessels covered by pericytes, the pericyte segmentation was used for filtering out blood vessel segments, that is, voxels in the medial axes of the blood vessels that were not inside the identified pericytes were removed. The sum of arc-lengths of the remaining medial axes divided by the total length of the blood vessels defined the fraction of blood vessels covered by pericytes.

Microglia were detected using a Laplacian of Gaussian blob segmentation approach[58]. Each 3D image was convolved with a Laplacian of Gaussian kernel having a standard deviation of 5 μm. Peaks of the resulting image having an intensity larger than 15 were associated with candidate microglia. Each peak was used for defining spheres representing microglia with an estimated radius of $5\sqrt{3}$ (Supplementary Fig. 1b). If two spheres overlapped by more than 0.2, one of them was randomly chosen and removed from the image. The parameters of the microglia segmentation were chosen to maximise the f1 score[59] of the result. To quantify microglia-vessel interaction, the distances between the detected microglia and the segmented blood vessels were calculated and the average value was taken. Also, microglia located less than 5 μm apart from a blood vessel were considered to contact the blood vessel.

The computational analyses were implemented using customised routines developed in the Python language.

**Ultrastructure.** Two acrolein/PFA-perfused brain sections containing the barrel cortex (Bregma −1.67 mm[60]) were selected per animal and processed for scanning electron microscopy, as previously described[21]Sections were washed in PBS, quenched in 0.3% $H_2O_2$ (in PBS) for 10 min, washed, permeabilized in 0.1% $NaBH_4$ (in PBS) for 30 min, and washed again. After quenching and permeabilization, sections were placed for 1 h at RT in blocking buffer (10% foetal bovine serum, 3% bovine serum albumin, 0.01% Triton X-100 in 50 mM TBS, pH = 7.6) and incubated overnight at 4 °C with the primary antibody rabbit anti-Iba1 (1:1000 in BB; cat# 019-19741, FUJIFILM Wako Chemical, Osaka, Japan). The next day, brains sections were washed in Tris-buffered saline (TBS), incubated for 1.5 h at RT with the secondary antibody biotinylated goat anti-rabbit (1:300 in TBS; cat# 111-066-046, Jackson ImmunoResearch, West Grove, PA, United States), followed by the avidin-biotin complex (1:1:100 in TBS; cat# PK-6100,Vector Laboratories, Burlingame, CA, United States) for 1 h at RT, then washed and revealed in 0.05% diaminobenzidine (DAB, 0.015% $H_2O_2$, in TBS; cat# D5905-50TAB, MilliporeSigma). Sections were next processed for electron microscopy (EM). Tissues were incubated in 3% ferrocyanide (in $H_2O$; cat# PFC232.250, BioShop, Burlington, ON, Canada) combined (1:1) with 4% aqueous osmium tetroxide (cat# 19170, Electron Microscopy Sciences, Hatfield, PA, United States) for 1 h, washed in PBS, incubated in 1% thiocarbohydrazide (in PBS; cat# 2231-57-4, Electron Microscopy Sciences) for 20 min, washed in PBS, incubated in 2% osmium tetroxide (in $H_2O$), then dehydrated in ascending concentration of ethanol (35%, 50%, 70%, 80%, 90%, and 3 times in 100%) followed by incubation in propylene oxide. Post-fixed sections were embedded in Durcupan ACM resin (cat# 44611–44614, MilliporeSigma) for 24 h, placed between two ACLAR® embedding sheets (cat# 50425-25, Electron Microscopy Sciences) and resin was polymerised at 55 °C for 72 h. Region of interest—the barrel cortex—was excised, glued on a resin block, and cut into 75 nm thick ultrathin sections using a Leica Ultracut UC7 ultramicrotome (Leica Biosystems).

Ultrathin processed sections were collected on a silicon nitride chip, glued on specimen mounts, and imaged at 5 nm resolution $(x, y)$ using a Crossbeam 540 Gemini scanning electron microscope (Zeiss), operating with an acceleration voltage of 1.4 kV and current of 1.2 nA. Analyses of the micrographs were performed by an experimenter blind to the conditions using ImageJ software. Details of distinguishable characteristics of the different organelles, cell types and parameters are provided in the Supplementary information (see Supplementary Fig. 1c, Supplementary Fig. 7 for additional examples of parameters analysed).

For the ultrastructure analysis of barrel cortex capillaries, area and perimeter of the endothelium were traced using the "freehand" tool and measured. The numbers of total mitochondria, mitochondria in fusion/fission stage (showing a "8" shape) and tight junctions were quantified per endothelium (Supplementary Fig. 7a, b). For each straight tight junctions with an angle between 15 and 80° (measured from the basement membrane), length was measured using the "line" tool and thickness was averaged by measuring the length between the tight junction and basement membrane using the "straight line" tool at three equidistant points along the tight junction. The average thickness of the basement membrane was determined by measuring the length of the basement membrane at four coordinates with the "straight line" tool (Supplementary Fig. 1c). The perimeter of the basement membrane was also traced using the "freehand" tool to evaluate astrocytic endfeet coverage. A number of other perivascular cells processes (pericytes and other) were also quantified for each capillary and the presence of lipidic inclusions was noted (Supplementary Fig. 7a, c–e). Pericytic processes coverage was calculated by dividing the total number of pericytic processes by the perimeter of the endothelium. Finally, astrocytic endfeet were quantified to measure the astrocytic coverage (number of endfeet divided by the perimeter of basement membrane), and traced using the "freehand" tool to determine the average endfeet size. Accumulation of glycogen granules was noted when observed within astrocytic endfeet.

For microglia ultrastructure analyses, organelles (i.e., endosomes, endoplasmic reticulum, Golgi apparatus, lipid bodies, lysosomes, lipofuscins, and mitochondria) and their anomalies (i.e., dilation of endoplasmic reticulum and Golgi apparatus cisternae)[61,62] within microglial cell bodies were quantified on a cellular basis using the ImageJ "counter" plugin. Number of microglial cell bodies contacting other cells (i.e., neurons, astrocytes, dark processes, microglia, oligodendrocytes; Supplementary Fig. 7f–i), neuronal elements (i.e., myelinated axons and synapses—presynaptic axon terminals and postsynaptic dendritic spines; Supplementary Fig. 1c), extracellular digestion activities (degenerating myelin, digestive exophagy; Supplementary Fig. 7j), blood vessels (Supplementary Fig. 1c), and extracellular space pockets were also quantified on a cellular basis as previously described by our team[18,61–65].

**Ultrastructural parameters recognition.** Dilation of the endoplasmic reticulum and Golgi apparatus cisternae was identified when the space between both cisternae

membranes was greater than 50 nm[1]. Lysosomes were distinguished by their heterogenous electron-dense round structure, which was subdivided into primary, secondary (when associated with endosomes) and tertiary (when associated with lipidic inclusions and often with endosomes) categories[2,3]. Lipidic inclusions regrouped lipofuscins as well as lipidic bodies. Lipofuscins were recognised by their electron-dense spherical structure with a distinct fingerprint-like pattern, whereas lipid bodies showed a smooth and uniform pattern[3]. Mitochondrial elongation was identified by a length of 1 μm or greater[1].

Microglia were recognised by their immunoreactivity to Iba1 as well as ultrastructural features including: their dark irregular nuclei with a heterogenous chromatin pattern and dark irregular cytoplasm, often containing long stretches of endoplasmic reticulum cisternae and lipidic inclusions (i.e., lipofuscins, lipid bodies, lipid droplets and lysosomes)[4]. Neurons were distinguished by their pale nuclei and pale cytoplasm, often with an apical dendrite and synaptic contacts[4] (Supplementary Fig. 7f). Synapses were identified by a visible synaptic density between the presynaptic axon terminal containing synaptic vesicles and the postsynaptic dendritic spine[4] (Supplementary Fig. 1c). Astrocytic cells were identified by their pale nucleus with a thin rim of heterochromatin and their pale irregular cytoplasm, often containing intermediate filaments[4]. Glycogen granules were recognised as dark electron-dense granules[4] (Supplementary Fig. 7g). Oligodendrocytes were distinguished by their dark round or oval nuclei with a heterogenous chromatin pattern and their dark squared-shape wide cytoplasm containing short, wide endoplasmic reticulum cisternae and often enriched in ribosomes[4] (Supplementary Fig. 7h). Capillaries were counted when microglial cell bodies directly touched their basement membrane (Supplementary Fig. 1c), which forms a thin electron-dense layer encompassing the capillary's cells including endothelial cells, pericytes and other perivascular cells (Supplementary Fig. 7a). Digestive exophagy, also known as extracellular digestion, was identified by extracellular space pockets containing degraded elements or debris in areas directly adjacent to the microglial cell body[5–7] (Supplementary Fig. 7j). Degraded myelin was recognised by ballooning, swelling or distancing between the well-defined myelin sheaths[4] and was often observed with exophagy (Supplementary Fig. 7j).

**Transcriptomics:Whole-transcript expression analysis by Clariom^TM S**. At P30, flash-frozen cortex of each offspring ($n = 4$ animals/diet) were homogenised in Trizol (cat#15596-026, Ambion, Austin, TX, United States) and total RNA was extracted using the Trizol/chloroform method followed by an isopropanol precipitation. The RNA pellet was washed once in 75% ethanol, let dry, then reconstituted in Nuclease-free water (cat#AM9937, Ambion). After solubilisation in RNAse-free water, RNA was passed through high Pure PCR Cleanup micro kit following the manufacturer's guidelines (cat#04983912001, Roche, Basel, Switzerland) to eliminate potential contaminants.

Total RNA samples were sent to Genome-Québec for whole transcript expression analysis. Sample quantity was measured by NanoDrop Spectrophotometer ND-1000 (NanoDrop Technologies LLC., Wilmington, DE, United States) and integrity was evaluated using a 2100 Bioanalyzer (Agilent Technologies, Santa Clara, CA, United States). Sense-strand cDNA was synthesised from 100 ng of total RNA. According to the manufacturer's instructions (Thermo Fisher Scientific), ssDNA were produced by fragmentation and labelling using GeneChip® whole-transcriptome terminal labelling kit. 3.5 μg of cDNA was hybridized on Clariom^TM S mouse array (Thermo Fisher Scientific) for 17 h at 60 rpm. Clariom^TM S mouse arrays were washed with GeneChip® Fluidics Station 450 (Thermo Fisher Scientific) using the GeneChip Hybridization Wash and Stain kit. Finally, microarrays were scanned on GeneChip® scanner 3000 (Thermo Fisher Scientific) and signal was analysed for the whole mouse genome using the Transcript Analysis Console (TAC) 4.0 software, generating differentially expressed gene lists, where threshold fold change was ±2.0 (Applied Biosystems, Thermo Fisher Scientific).

Data of the whole transcript expression experiment is accessible on an opensource platform (FigShare, 16786504).

**FISH-immunofluorescence analysis**. RNAScope Multiple Fluorescent Assay was done using brain sections ($n = 6$ animals/diet for which 3 were males and 3 were females, 10–12 sections per animals) and samples were prepared for RNAScope assay following the manufacturer's protocol: "Sample Preparation Technical Note for Fixed Frozen Tissue Using RNAscope® Fluorescent Assay" (technical note: 320535 Rev A; Advanced Cell Diagnostics; Newark, CA, USA). Briefly, brain sections were subjected to antigen retrieval at 100 °C for 5 min in manufacturer's Target Retrieval solution (cat#322000; Advanced Cell Diagnostics; Newark, CA, USA). Following antigen retrieval step, sections were exposed to Protease III digestion (cat# 322340; Advanced Cell Diagnostics; Newark, CA, USA) for 30 min at 40 °C.

After the sample preparation step, labelling of target RNA was performed by exactly following the manufacturer's protocol: "RNAscope® Fluorescent Multiplex Kit User Manual PART 2" (document number: 320293; Advanced Cell Diagnostics; Newark, CA, USA). The following fluorescently labelled target probe C1: Mm-Igtp (Atto 550) was constructed by the manufacturer and provided in the RNAscope® Fluorescent Multiplex Detection Reagents Kit (cat#320851; Advanced Cell Diagnostics; Newark, CA, USA). After labelling and amplification steps, the slides were washed four times, for 5 min, in Wash Buffer (cat# 310091; Advanced Cell

Diagnostics; Newark, CA, USA) and immediately processed for immunofluorescent labelling of microglia.

Microglia were immunofluorescently labelled using the following procedure. Slides were rinsed three times in PBS for 5 min followed by two 5 min washes in 0.5% PBT (0.5% Triton X-100 in 50 mM PBS). Sections were then incubated for 1.5 h at RT in blocking solution (10% donkey serum, 0.5% cold water fish skin gelatin in 0.5% PBT). After blocking step, sections were incubated with rabbit α-Iba1 (1:400, cat# 019-19741, FUJIFILM Wako Chemicals, Osaka, Japan) in blocking solution overnight at 4 °C. Slides were rinsed in 0.5% PBT three times for 5 min and incubated with donkey α-rabbit A488 (1:300, Thermo Fisher Scientific, ON, Canada) in blocking solution for 2 h. Slides were subjected to two washes, 5 min, in 0.5% PBT followed by two 5 min washes in 100 mM PB. Slides were quickly dipped in distilled water and mounted with either Fluoromount-G or ProLongGold with DAPI.

For microglia expression analysis, immunostained sections were examined as previously described[3]. Briefly, a Zeiss Axio Imager M2 microscope equipped with an Axiocam 506 mono digital camera and a Zeiss ApoTome.2 module was used for optical sectioning at 20x magnification. The total optical thickness of 10 μm, step size of 1 μm, for a total number of 11 optical slices, was used to acquire images. Images were z-stack projected at maximum intensity using ImageJ and analysed as follow.

Complete cells co-labelled for Iba1$^+$ and DAPI$^+$ were first identified. Of these cells, the ones expressing at least 2 identifiers of the RNA of interest, Igtp, were considered as positive. From this definition, percentage of microglia that express the RNA of interest was calculated for each tissue section then averaged for each individual mouse.

**Behaviours**. At adulthood (P60-P85), offspring underwent behavioural assessment, from the least to most stressful test, and with 2 days of rest in between. A thorough characterisation of the functional outcomes including a vast range of behaviours known to be altered in neurodevelopmental disorders (e.g., motor, social, cognitive, mood, sensorial) in the animal life course, including adulthood[66–70], was performed. General motricity was assessed by open field test. Social preference and social novelty preference were measured by three-chambers social interactions. Novel object recognition was used to evaluate spatial memory. Marble-burying test was used to assess repetitive and stereotypic behaviours. Anxiety under normal conditions was gauged using the open field and elevated plus maze tests. Sensorimotor gating of the acoustic reflex was measured by the prepulse inhibition test. Detailed behaviour methods, excluding marble burying, can be found in the Supplemental behaviour information.

For each behavioural test, animals were acclimated to the experimental room for 30 min prior to testing under the experimental lighting condition. All behaviours were assessed in a way that prevented mice from seeing the experimenter during testing. Between each trial and test, arena and objects were thoroughly cleaned with ethanol/peroxide mixture to prevent any olfactory cues. Except for the open field, marbles burying and prepulse inhibition (PPI), all behavioural tests were filmed and scored automatically by the tracking software TopScan Version 2.00 (Clever Sys Inc, Reston, VA, United States), and scoring was verified by an observer blinded to experimental conditions.

*Marble burying*. Under normal lighting conditions (30 lux) at the beginning of light phase (between 8:00 and 11:00), the mice were placed in a transparent Plexiglas cage (17 cm (width) × 30 cm (length) × 13 cm (height)) filled with ~2 cm-thickness of wood chip and 12 round marbles (1 cm (diameter); 3 marbles × 4 marbles) distanced ~2 cm from each other. To prevent the animal from escaping, a cap was placed on top of the cage for the whole duration of the test. After 30 min, the mice were removed, and number of marbles buried up to 2/3 was counted independently by three different double-blind observers to prevent any subjective bias.

*Open field*. Mice were placed in a transparent Plexiglas open field (40 cm (width) × 40 cm (length) × 30 cm (height)) connected to the VersaMax system (*Accusan Instruments Inc.*, Columbus, OH, United States) for 15 min after the dark phase (between 8:00 and 11:00). The test was divided into first 5 min of habituation and 10 min of testing. VersaMax tracking system automatically recorded global locomotion parameters: speed, distance travelled, time spend in different areas of the arena (i.e., centre *versus* edge *versus* corner), horizontal and vertical movements, clockwise and anti-clockwise rotations, as well as stereotypic behaviours.

*Novel object recognition*. Novel object recognition consisted of a training and a testing phase in an opaque plexiglass box (40 cm (width) × 40 cm (length) × 30 cm (height)) under the infrared light in the beginning of light phase (between 8:00 and 11:00). Mice were, first, let to explore two identical objects for 5 min during the acquisition phase. Then, the animals were put back in their home cage for 30 min. During the novel object recognition retention phase, mice were placed in the same experimental arena for 5 min in the presence of one of the two familiar objects along with a novel object of similar size. The novel object was randomly assigned. Object exploration was defined as touching or sniffing within the interaction zone (within ~2 cm of the object). The recognition index was calculated by the ratio of the time spent exploring the object over the total time allowed. Mice with an exploration time lower than 10 s per object were considered unsuitable and were not used.

*Three-chambers social interaction.* For two days prior to testing, stranger mice were habituated to the wired cage under infrared light for 20 min to minimise animal stress during the social interaction assessment. Experimental mice were placed for 10 min in the central compartment and let free to explore the 3-chambers compartment (chamber: 26 cm (length) 21.6 (width) × 21.6 (height); door: 5 cm (width) × 5 cm (height)) with the two wired cages in the two side compartments. After the habituation to the three chambers, a stranger mouse was placed in a cylindrical wired-cup (dimension: 7.6 cm (D) × 9.5 cm (H)) in one of the two side compartments and a toy in the other one. Social preference was assessed for 10 min, then the toy was substituted with a novel stranger to evaluate social novelty preference for another 10 min session. Time spent and number of entries in each compartment, as well as contacts/sniffing time with the occupied wired cage were measured using TopScan software. Social preference index and social novelty index were respectively calculated by the ratio of time interacting with stranger over toy, and time interacting with novel over familiar stranger.

*Elevated plus maze.* Elevated plus maze was assessed at the beginning of the light phase (8:00–11:00) under normal lighting (~30 lux). Mice were placed at the junction of the open and closed arms facing the open arms and let free to explore the elevated plus maze (open arm: 29.25 cm (length) × 5 cm (width); closed arm: 29.25 cm (length) × 5 cm (width) × 11.5 cm (height); centre: 5 cm (width) × 5 cm (length)) for 10 min. After the 5 min session, animal was put back in its home cage. Time spent and number of entries in open arms, closed arms and at the centre were measured to evaluate anxiety-related behaviour at basal level using TopScan software. Entry was considered when all three paws entered the arm.

*PPI.* PPI is the inhibition of the startle response to a strong auditory stimulus when the stimulus is preceded by a weaker stimulus[8]. Alterations of the PPI are known to be commonly observed in neurodevelopmental disorders. We measured the sensorimotor gating which is altered in neurodevelopmental disorders, such as schizophrenia[8]. Animals were placed in a cylindrical Plexiglas animal enclosure inside a commercially available system (SR-LAB; San Diego Instruments, San Diego, CA, United States). After 5 min of acclimation inside the enclosure, 42 discrete trials were performed; the first two trials are 120 dB in magnitude followed by 40 trials in which the startle pulse was either alone or 100 ms prior to a 30 ms prepulse. The prepulses of intensity ranging from 3 to 15 dB were randomly presented to experimental mice. Startle responses were determined automatically by the SR-LAB system. PPI was calculated as the relative percentage of the mean amplitude of the startle response without prepulse compared to those recorded following a prepulse.

**Statistics and reproducibility**. All statistical analyses were conducted using Prism 8 (v.8.4.2, GraphPad Software, San Diego, CA, United States) and results presented as whisker box graphs or violin plot showing median, min and max. Sample size was defined by the number of individual animals for all analyses except for ultrastructure analyses in which sample size represented the number of individual microglial cells or capillaries. Normality was checked with Shapiro–Wilk test. For normally distributed dataset, Grubbs' test was used to remove outliers from the datasets prior to parametric statistical analysis. To reveal variance difference between diet (CD vs mHFD) and with interaction with the sex (male vs female), two-way ANOVAs were used for normally distributed dataset, while mixed-effect models were used for non-normally distributed dataset (i.e., occipital cortex vessels length, hippocampus density ratio) when testing for Diet and Sex main effects as well as Diet*Sex interaction effects (Supplementary Tables 1–3). Significant Diet*Sex interactions were followed by post-hoc Bonferroni to identify significantly different groups (CD-male vs CD-female vs mHFD-male vs mHFD-female). For transcriptomic analyses, differential expression was tested by empirical Bayes statistical analysis comparing CD vs mHFD, while for FISH-immunofluorescence analysis differences were tested by unpaired Student's *t* test for data set following a normal distribution. Significance was reached when *P* value was 0.05 or lower.

**Reporting summary**. Further information on research design is available in the Nature Research Reporting Summary linked to this article.

## Data availability

Source data for ClariomTMS experiments are available (https://doi.org/10.6084/m9.figshare.16786504). All other data and protocols are available from the corresponding authors upon request.

## Code availability

The custom scripts for vascular and microglial quantifications, written in Python, are available on GitHub (https://github.com/chcomin/CommunicationsBiology2021) and from the Comin lab (chcomin@gmail.com).

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

## Acknowledgements

In memory of Giamal N. Luheshi who made possible the current study, allowing the people involved to meet, to discuss, and to move forward with the current study. We thank all the team of Dr. Baptiste Lacoste that took the time to share their expertise with M.B. during her training in his lab, as well as Genome-Québec for their service and support in performing the whole-transcript expression experiment by Clariom™ S. We also thank Iris Kim and J. Kasia Szyszkowicz for their technical support with animal handling. We also thank all the funding agencies that made this work possible. During this study, M.B., L.F.d.C., C.L. and F.G.I. were, respectively, recipient of the doctoral award from Fonds de recherche du Québec—Santé (FRQS), Vanier Canada Graduate scholarship from Canadian Institutes of Health Research (CIHR), the returning student award of the Faculty of Medicine (McGill University), and the full doctoral scholarship from CONACYT (Mexican Council of Science and Technology). This research study was funded by a NSERC Discovery grant awarded to M.E.T. (#RGPIN-2014-05308), by Conselho Nacional de Desenvolvimento Científico e Tecnológico (CNPq) awarded to L.d.F.C. (#307085/2018-0), by Fundação de Amparo à Pesquisa do Estado de São Paulo (FAPESP) awarded to C.H.C. (#18/09125-4) and to L.d.F.C. (#15/22308-2). M.E.T. is a Tier 2 Canada Research Chair in *Neurobiology of Aging and Cognition*.

## Author contributions

M.B., B.L. and M.E.T. conceived the project and designed experiments. Animal handling and tissues processing was performed by M.B., L.F.d.C. and C.L. M.B. and L.F.d.C. performed behavioural analyses. M.B. analysed 2D images of microglia as well as ultrastructure images of capillaries and microglia. Statistical analyses were run by M.B. except for the transcriptomics and the FISH-immunofluorescence experiments. C.H.C. and L.d.F.C. developed the 3D vessels quantification methods. C.H.C. performed the computational 3D analyses. M.F.A. optimised and performed FISH-immunofluorescence. M.B., M.F.A. and F.G.I., respectively, performed the imaging for 2D/3D analysis by epifluorescence microscopy, for FISH-immunofluorescence analysis by epifluorescence microscopy and for ultrastructure analysis by scanning electron microscopy. J.R.N. analysed FISH-immunofluorescence images and ran statistical analysis of the FISH-immunofluorescence data. Whole-transcript expression experiment was performed by Genome-Quebéc and analysed by M.W. M.B. designed and made figures and drawings used in this manuscript and modified upon suggestion of all co-authors. M.B., C.H.C., L.F.d.C., C.L., M.F.A., F.G.I., J.R.N., M.W., M.C., L.d.F.C., B.L. and M.E.T. contributed to writing, proofreading and approbation of the final manuscript.

## Competing interests

The authors declare no competing interest.
