## [Transparent Peer Review File · Communications Biology]

Reviewers' comments:

Reviewer #1 (Remarks to the Author):

Bordeleau et al. examine how maternal high-fat diet exposure during gestation impacts the offspring to result in cerebrovascular, microglial and behavioural alterations. The authors show that maternal high-fat diet exposure during gestation results in an increased vascular network density and microglial-blood vessel interactions in both male and female adolescent offspring. They also show that specific genes were altered in the offspring of high-fat diet exposed dams, and that marble burying was increased in these gestationally exposed offspring. While understanding how maternal high-fat diet exposure during gestation impacts neurovascular development is interesting and may involve alterations to microglial-blood vessel interactions, this study needs to address some gaps in the data in order to support their arguments and conclusions.

Major Concerns:

1. In general, the imaging quality throughout the manuscript is poor and should be improved. In Figure 2b-i the authors should change the colour of IBA1+ microglia from white to red and they need to provide super-resolution images showing the interactions between microglia and vessels. It would also be nice if the authors provided 3D reconstructions of the microglial-vessel interactions. Moreover, in Extended Figure 2a-d the authors need to provide super-resolution images showing pericyte coverage on the vessels, as the imaging quality is poor in this figure as well.

2. In the introduction, the authors say: "It has also been demonstrated that maternal obesity disturbs development of the blood-brain barrier in neonate rodent offspring¹¹. Furthermore, mHFD-mediated maladaptive brain development has been shown to increase vascular permeability in the hypothalamus of foetal, neonatal¹¹ and adult^{12,13} rodent offspring." However, in this study the authors show that maternal high-fat diet exposure during gestation results in increased vascular density and branching in the cortex, so it is important to understand if these additional vessels are defective in some manner. Moreover, considering that the authors state that: "While endothelial cells create the vascular wall^{7,15,16}, pericytes are mural cells required for formation and maintenance of brain blood vessels during embryonic development, formation and integrity of blood brain barrier, as well as regulation of cerebral blood flow during postnatal life⁷", but see no changes to pericytes, the authors need to thoroughly examine other aspects of the vasculature. For example, the authors need to examine tight junctions, endothelial cells and smooth muscle cells in their high-fat diet exposed offspring and compare these to controls.

3. The authors also need to examine how maternal high-fat diet exposure during gestation impacts astrocyte endfoot coverage of vessels, as Contu et al., 2019 showed that postnatal high-fat diet exposure results in a sex-specific loss of astrocyte endfoot coverage of arteries in female mice. Moreover, considering that Mondo et al., 2020 show that juxtavascular microglia associate with vascular areas void of astrocyte endfeet and that there is a developmental shift in microglial migratory behaviour along vessels which corresponds to where astrocyte endfeet more fully ensheath vessels, it is essential that the authors examine how maternal high-fat diet exposure during gestation impacts astrocyte endfoot coverage of vessels as the authors see a microglial-vessel effect.

4. On page 5 the authors say: "No interaction between mHFD-exposure and sex was detected statistically (Extended Figure 3), hence male and female data were pooled", yet the authors then go on to say that: "Moreover, regardless of maternal diet, cortical microglia from female offspring were closer to blood vessels compared to their male littermates, an effect observed throughout the cortex (Whole cortex: $p=0.003$; Anterior cortex: $p=0.006$; Parietal cortex: $p=0.003$; Occipital cortex: $p=0.009$; Extended Figure 3)." Measurements that show a sex effect should not be pooled, and ultimately, I am confused as to whether the data in Figure 2j-m is the same as the data in Extended Figure 3f-I, and the data in Figure 2o-r is the same as the data in Extended Figure 3j-m? If this is the

case, get rid of Figure 2 where males and females were pooled and make Extended Figure 3 a new Figure 2. Additionally, it looks as though the males are more susceptible in Extended Figure 1a-d: there is no significance? I wonder if given the points above, the males and females should be graphed separately in the text. Furthermore, the discussion of diet effects versus sex effects is very confusing throughout the text and should be addressed.

5. The authors state that: "Transcriptome analysis identified significantly decreased levels of 11 transcripts and significantly increased levels of four transcripts (Figure 3a). Out of these 15 transcripts, four belong to the haemoglobin pathway. Hba-a2; Hba-a1, Hbb-bs; Hbb-b1, Hbb-bt; Hbb-b2 were reduced by 3.19 to 5.58-fold, and Erdr1 increased by 16.48-fold, in the mHFD-exposed offspring (Figure 3b-c)" and that: "In endothelial cells, these factors are implicated in the regulation of nitric oxide signaling²¹ which regulates angiogenesis and vascular reactivity²²." This is further support for why the authors need to look at how maternal high-fat diet exposure impacts endothelial cells. ^[1]_{SEP}

6. During their presentation of the transcriptomic data (throughout page 7), the authors make statements such as: "Together with the alteration of Csprs and Igtp, these results highlight an immune dysregulation, in which microglia may actively participate, via functional alteration of their physiological roles in helping neurodevelopmental processes, such as angiogenesis, synaptic pruning and/or myelination^{9,18,20}". The authors are making these generalizations based on microarray technology. Therefore, the authors need to confirm these changes using tools such as RNAScope, where they can both look at the change in levels of specific mRNA transcripts and determine in which cell types these changes are occurring. ^[1]_{SEP}

Minor Concerns:

1. It would be interesting to see if similar to Contu et al., 2019 the authors find increased coverage of blood vessels by phagocytic CD68+/IBA1+ microglia.
2. It is unusual to include references in the abstract and unless the journal prefers they be kept, I suggest the references be removed.

Reviewer #2 (Remarks to the Author):

Bordeleau et al. investigated the effect of maternal high fat feeding on the NVU, microglial-NVU contact and behavioural outputs in young offspring. They found that vessel density, the number of vessel branching points, microglia-vessel contact and marble burying was higher in offspring born to HF-fed mother compared to those whose mothers were fed standard chow. Understanding the impact of maternal obesity on the structure and function of the brain has important implications for both neurodevelopmental and neurodegenerative diseases. As the authors state, less work has been done to specifically investigate the impact of maternal obesity on the NVU and therefore this is a topical paper. The statistical analysis is sound and in general, sufficient detail is given in the methods for the study to be reproduced (although previous experience with some of the behavioural tests would be needed to fully replicate those experiments). However, there are several major limitations to the current study that reduce its novelty and impact.

1. While the main thrust of the paper is to investigate the NVU, only a few aspects of the NVU were examined. For example, no analyses were carried out on the blood-brain barrier, astrocyte expression, smooth muscle cells or neuronal innervation. Analyses of vessel properties were grouped together, despite the functional differences between capillaries, arteries and veins. Therefore, potentially interesting effects of maternal obesity on other NVU components and/or relationships between microglia contact with specific vessel types that would have advanced the current understanding may

have been missed.

2. Changes in gene expression were not validated by any protein analysis and there was no exploration of some of the potential pathways that the authors suggest may underlie the observations in the offspring brains. For example, the authors suggest that changes in Hhb and/or Erdr1 expression may play a role in the increased vascular density observed in the HFD offspring via alterations in endothelial NOS expression. Why didn't they determine if eNOS expression was changed in the HFD mice? Such experiments could have additionally helped determine cell specificity of gene expression which couldn't be achieved using whole-cell transcriptomics.

3. Two different offspring ages were used for histology and behavioural analyses (P30 and P60-P85, respectively). What is the rationale? Why weren't the histology experiments carried out on the brains of the animals either at the same age as those tested or following completion of the behavioural task, so that a more direct inference could be made between NVU/microglial changes and behavioural consequences? How do the authors know that the histological changes observed at P30 persisted at later ages? Moreover, how can the authors distinguish behavioural changes induced by alterations in vascular or microglial properties from those that may arise independently due to maternal HF-induced neuronal dysfunction?

4. A significant amount of work was done on microglia, but there was very little discussion of the current work in the context of previous studies looking at maternal HF and brain inflammation/microglia, despite the fact that the authors themselves have published on this topic (Bordeleau et al. *J Neuroinflammation*. 2020; 17: 264; Bilbo and Tsang *FASEB J* 2010 Jun;24(6):2104-15; Smith et al. *Brain, Behaviour and Immunity* 2020, 84:80-89). There was also no discussion of the observed regional variation in the effect of maternal HF on vessel density or microglial contact or discussion of why male and female offspring were equally affected by maternal HF when several other studies have suggested sex-specific susceptibility to prenatal HF exposure (for example, Edlow et al. *Am J Obstet Gynecol* 2016, 214, 623 e621-623 e610; Glendining and Jasoni *Int J Mol Sci* 2019;20(2); Vucetic et al. *Endocrinology* 2010 151(10), 4756-4764).

5. More complete information is needed about the diet compositions (e.g. AFE from fat, protein, carbohydrates). Were the chow and HF diet otherwise matched for vitamins and other components that affect brain development (e.g. folic acid, iron etc)?

Minor comments: The manuscript should be reviewed for typos (e.g. Fig. 3, extended Fig. 5a) and formatting (e.g. Fig. 4).

Reviewer #3 (Remarks to the Author):

In this manuscript, the authors investigated the effects of maternal HFD on neurovascular development, with a particular focus on microglial interactions with the blood vessels, as well as on a range of adult behaviours. This is an interesting study, integrating a range of novel methodologies with respect to the analysis of immunohistochemical data. I have, however, some major concerns, particularly regarding the statistical analyses.

Specific comments:

What was the rationale for choosing the adolescence as the first timepoint of observation? How does this time correspond to the developmental trajectory of the neurovascular system?

Please provide more details about animal use. How many dams in total contributed to this study? Were there any effects of HFD on pregnancy and litter outcomes? Including body weight at birth, pup mortality and sex ratio?

The decision to pool the data when the effect of sex was not significant is questionable. It is equally important to show the direction of change is similar in males and females. If the study is sufficiently powered, then the effects of treatment should be similar in each sex alone, as when the data are pooled and the n is larger. Please present all data in the same manner, with sex as an independent variable.

The results of the transcriptome analysis are unclear. Differences were detected in 15 transcripts, but out of how many in total? Can the authors include more detail about this array, including the analysis strategy (threshold fold change), and supplementary tables presenting changes in other genes that were detected in the array.

What was the rationale to conduct all the different behavioural tests in this study? What is the role of neurovascular function in underlying the development of these behaviours? Were all behaviours conducted in the same set of animals? What was the order of the tests? The authors mention neurodevelopmental disorders as the precedence for behavioural testing. However, the tests were conducted in adulthood and not during early development, when neurodevelopmental disorders are likely to emerge.

Overall, out of the 6 tests, with each test providing diverse information about a range of behaviours, differences were detected only in the marble burying task. These data suggest there were overall minimal long-term effects of maternal HFD, despite changes in neurovascular parameters in adolescence. I therefore suggest the authors to rewrite their conclusions emphasising these encouraging findings, rather than suggesting the effects of maternal HFD were detrimental.

The authors incorporate interesting microglial analyses in this study. Could the authors indicate whether in addition to changes in microglia-blood vessel proximity were there also changes in microglial numbers? No changes in microglial density suggest there were no difference in their activation state, but could the authors elaborate about this in the manuscript.

Language editing is recommended.

Reviewers' comments

Reviewer #1 (Remarks to the Author):

Bordeleau et al. examine how maternal high-fat diet exposure during gestation impacts the offspring to result in cerebrovascular, microglial and behavioural alterations. The authors show that maternal high-fat diet exposure during gestation results in an increased vascular network density and microglial-blood vessel interactions in both male and female adolescent offspring. They also show that specific genes were altered in the offspring of high-fat diet exposed dams, and that marble burying was increased in these gestationally exposed offspring. While understanding how maternal high-fat diet exposure during gestation impacts neurovascular development is interesting and may involve alterations to microglial-blood vessel interactions, this study needs to address some gaps in the data in order to support their arguments and conclusions.

Major Concerns:

1. In general, the imaging quality throughout the manuscript is poor and should be improved. In Figure 2b-i the authors should change the colour of IBA1+ microglia from white to red and they need to provide super-resolution images showing the interactions between microglia and vessels. It would also be nice if the authors provided 3D reconstructions of the microglial-vessel interactions. Moreover, in Extended Figure 2a-d the authors need to provide super-resolution images showing pericyte coverage on the vessels, as the imaging quality is poor in this figure as well.

Thank you for your suggestions. We have accordingly improved the resolution and quality of the images throughout the manuscript. However, considering that the pericytes were coloured in red throughout the representative 3-D reconstruction examples (Supplementary Figure 1b), we opted to keep Iba1⁺ microglial cells in white (corresponding to far-red) to preserve formatting uniformity across figures and prevent possible confusion by the readers. We also now include several examples showing at nanoscale-resolution the ultrastructure of the different cell types of the neurovascular unit (Figures 2-3, Supplementary Figures 1-3,5,7, Supplementary Table 2-3).

2. In the introduction, the authors say: "It has also been demonstrated that maternal obesity disturbs development of the blood-brain barrier in neonate rodent offspring¹¹. Furthermore, mHFD-mediated maladaptive brain development has been shown to increase vascular permeability in the hypothalamus of foetal, neonatal¹¹ and adult^{12,13} rodent offspring." However, in this study the authors show that maternal high-fat diet exposure during gestation results in increased vascular density and branching in the cortex, so it is important to understand if these additional vessels are defective in some manner.

Moreover, considering that the authors state that: "While endothelial cells create the vascular wall^{7,15,16}, pericytes are mural cells required for formation and maintenance of brain blood vessels during embryonic development, formation and integrity of blood brain barrier, as well as regulation of cerebral blood flow during postnatal life⁷", but see no changes to pericytes, the authors need to thoroughly examine other aspects of the vasculature. For example, the authors need to examine tight junctions, endothelial cells and smooth muscle cells in their high-fat diet exposed offspring and compare these to controls.

Thank you for your advice, we now have provided complementary nanoscale super-resolution characterization of the neurovascular unit using scanning electron microscopy, in which we examined changes in endothelial cells (area, number of mitochondria and tight junctions), pericyte coverage, basal membrane thickness as well as microglial and astrocytic endfeet coverage of brain capillaries.

These results are now presented in Figures 2-3 and Supplementary Figures 2,5, as well as Supplementary Tables 2-3.

3. The authors also need to examine how maternal high-fat diet exposure during gestation impacts astrocyte endfoot coverage of vessels, as Contu et al., 2019 showed that postnatal high-fat diet exposure results in a sex-specific loss of astrocyte endfoot coverage of arteries in female mice. Moreover, considering that Mondo et al., 2020 show that juxtavascular microglia associate with vascular areas void of astrocyte endfeet and that there is a developmental shift in microglial migratory behaviour along vessels which corresponds to where astrocyte endfeet more fully ensheath vessels, it is essential that the authors examine how maternal high-fat diet exposure during gestation impacts astrocyte endfoot coverage of vessels as the authors see a microglial-vessel effect. Thank you. We have now quantified ultrastructural interactions of astrocytes and microglia with the vasculature. Astrocytic endfeet had similar size and coverage. Although 3D-reconstruction analysis identified an increased proximity of microglia to blood vessels, microglial cell bodies made less direct contacts with the basement membranes of capillaries.

These results are now presented in Figures 2-3 and Supplementary Tables 2-3.

4. On page 5 the authors say: “No interaction between mHFD-exposure and sex was detected statistically (Extended Figure 3), hence male and female data were pooled”, yet the authors then go on to say that: “Moreover, regardless of maternal diet, cortical microglia from female offspring were closer to blood vessels compared to their male littermates, an effect observed throughout the cortex (Whole cortex: $p=0.003$; Anterior cortex: $p=0.006$; Parietal cortex: $p=0.003$; Occipital cortex: $p=0.009$; Extended Figure 3).” Measurements that show a sex effect should not be pooled, and ultimately, I am confused as to whether the data in Figure 2j-m is the same as the data in Extended Figure 3f-l, and the data in Figure 2o-r is the same as the data in Extended Figure 3j-m? If this is the case, get rid of Figure 2 where males and females were pooled and make Extended Figure 3 a new Figure 2. Additionally, it looks as though the males are more susceptible in Extended Figure 1a-d: there is no significance? I wonder if given the points above, the males and females should be graphed separately in the text. Furthermore, the discussion of diet effects versus sex effects is very confusing throughout the text and should be addressed.

We apologize for the confusion. We now present all data split by sex throughout the manuscript. Additionally, coming from the same litters, male and female groups are part of the same dataset, which then are more appropriate to analyze together, while evaluating afterwards the potential effect of biological sex (*Sex* main effect and *Diet*sex* interaction effect) on the offspring brain. We carefully revised the manuscript and better clarified the discussion of main *Sex* main effect.

5. The authors state that: “Transcriptome analysis identified significantly decreased levels of 11 transcripts and significantly increased levels of four transcripts (Figure 3a). Out of these 15 transcripts, four belong to the haemoglobin pathway. Hba-a2; Hba-a1, Hbb-bs; Hbb-b1, Hbb-bt; Hbb-b2 were reduced by 3.19 to 5.58-fold, and Erdr1 increased by 16.48-fold, in the mHFD-exposed offspring (Figure 3b-c)” and that: “In endothelial cells, these factors are implicated in the regulation of nitric oxide signaling²¹ which regulates angiogenesis and vascular reactivity²².” This is further support for why the authors need to look at how maternal high-fat diet exposure impacts endothelial cells.

As per the reviewer suggestion, we have now included ultrastructural analysis by scanning electron microscopy that allowed us to directly measure at nanoscale super-resolution changes in endothelial cells, notably to gain further insight into metabolic shifts (mitochondria density as well as ratio of

mitochondrial fusion/fission). Furthermore, we quantified ultrastructural metabolic markers within astrocytic endfeet (glycogen granules accumulation), pericytes (number of lipidic inclusion) and microglial cell bodies (mitochondrial number and alterations, dilation of the endoplasmic reticulum cisternae, and number of lipidic inclusions).

These results are now shown in Figure 2-3, Supplementary Figures 2-3,5 and Supplementary Tables 2-3.

6. During their presentation of the transcriptomic data (throughout page 7), the authors make statements such as: “Together with the alteration of *Csprs* and *Igtp*, these results highlight an immune dysregulation, in which microglia may actively participate, via functional alteration of their physiological roles in helping neurodevelopmental processes, such as angiogenesis, synaptic pruning and/or myelination^{9,18,20}”. The authors are making these generalizations based on microarray technology. Therefore, the authors need to confirm these changes using tools such as RNAScope, where they can both look at the change in levels of specific mRNA transcripts and determine in which cell types these changes are occurring.

Thank you for your suggestion. We have now confirmed Iba1⁺ microglial expression of immune-modulating genes (*Csprs* and *Igtp*) using RNAScope technology, then quantified RNA expression changes of *Igtp*, coding for an immune-regulatory GTPase, which was the most impacted gene identified by microarray analysis (Figure 4). This approach allowed us to directly visualize *in situ* the expression of *Igtp* by microglia, which were found to be 80-83% positive, supporting microglial involvement in the immune and metabolic outcomes on neurovascular unit cells.

Furthermore, we also assessed changes in the ultrastructural interactions of microglial cell bodies with myelinated axons and synapses to provide insights into their involvement in synaptic pruning and myelination processes. These new results are shown in Figure 3.

Minor Concerns:

1. It would be interesting to see if similar to Contu et al., 2019 the authors find increased coverage of blood vessels by phagocytic CD68+/IBA1+ microglia.

Thank you. Although we did not specifically examine microglial expression of CD68—a marker of phagolysosomal activity, we further quantified direct cell-cell interactions of microglia with brain capillaries as well as their changes in organelles involved in the phagolysosomal pathway (immature lysosomes, mature lysosomes, endosomes) and markers of other phagocytic events (digestive exophagy and myelin degradation). (Figure 3, Supplementary Figure 5, Supplementary Table 3) Our analyses revealed no significant change in the phagolysosomal pathway among the overall microglial population except for sex differences where female compared to male microglia had more tertiary lysosomes and empty endosomes (Supplementary Table 4).

2. It is unusual to include references in the abstract and unless the journal prefers they be kept, I suggest the references be removed.

We apologize for the uncommon formatting. This was specific to the short communication format of another journal from the Nature series where our research paper was first submitted before its transfer to *Communications Biology*. We have now fully reformatted our manuscript to follow the formatting standards of *Communications Biology*.

Reviewer #2 (Remarks to the Author):

Bordeleau et al. investigated the effect of maternal high fat feeding on the NVU, microglial-NVU contact and behavioural outputs in young offspring. They found that vessel density, the number of vessels branching points, microglia-vessel contacts and marble burying was higher in offspring born to HF-fed mother compared to those whose mothers were fed standard chow. Understanding the impact of maternal obesity on the structure and function of the brain has important implications for both neurodevelopmental and neurodegenerative diseases. As the authors state, less work has been done to specifically investigate the impact of maternal obesity on the NVU and therefore this is a topical paper. The statistical analysis is sound and in general, sufficient detail is given in the methods for the study to be reproduced (although previous experience with some of the behavioural tests would be needed to fully replicate those experiments). However, there are several major limitations to the current study that reduce its novelty and impact.

1. While the main thrust of the paper is to investigate the NVU, only a few aspects of the NVU were examined. For example, no analyses were carried out on the blood-brain barrier, astrocyte expression, smooth muscle cells or neuronal innervation. Analyses of vessel properties were grouped together, despite the functional differences between capillaries, arteries and veins. Therefore, potentially interesting effects of maternal obesity on other NVU components and/or relationships between microglia contact with specific vessel types that would have advanced the current understanding may have been missed.

Thank you for your advice. We have now further examined cortical capillaries at nanoscale super-resolution using scanning electron microscopy and quantified changes in the neurovascular unit ultrastructure, specifically in endothelial cells (area, mitochondria density, rate of mitochondrial fusion/fission, number of tight junctions, tight junctions length and thickness, basement membrane thickness), pericyte (coverage and number of lipidic inclusions), astrocytic endfeet (coverage, size and presence of glycogen granules), and microglia [interaction with their environment (capillaries, synaptic elements, myelinated axons, neurons, astrocytes, oligodendrocytes, extracellular space pockets) and organelle content (number of lysosomes, endosomes, lipidic inclusions, mitochondria and their alterations, presence of endoplasmic reticulum and Golgi apparatus cisternae dilation)].

2. Changes in gene expression were not validated by any protein analysis and there was no exploration of some on the potential pathways that the authors suggest may underlie the observations in the offspring brains. For example, the authors suggest that changes in Hhb and/or Erdr1 expression may play a role in the increased vascular density observed in the HFD offspring via alterations in endothelial NOS expression. Why didn't they determine if eNOS expression was changed in the HFD mice? Such experiments could have additionally helped determine cell specificity of gene expression which couldn't be achieved using whole-cell transcriptomics.

Thank you for raising this concern. We have now confirmed Iba1⁺ microglial expression of immune-modulating genes (*Csprs* and *Igtp*) using RNAscope technology, then quantified RNA expression changes of *Igtp*, coding for an immune-regulatory GTPase, which was the most impacted gene identified by microarray analysis (Figure 4). This approach allowed us to directly visualize *in situ* the expression of *Igtp* by microglia, which were found to be 80-83% positive, supporting microglial involvement in the immune and metabolic outcomes on neurovascular unit cells.

To provide better understanding of whole-cortex transcriptomics data, we also quantified as mentioned above immune and metabolic ultrastructural alterations in neurovascular unit cells from

the cortex; looking at the endothelial cell astrocytic endfeet, pericytes and microglial cell bodies (Figures 2-3, Supplementary Figures 1,2,5,7 and Supplementary Tables 2-3).

3. Two different offspring ages were used for histology and behavioural analyses (P30 and P60-P85, respectively). What is the rationale? Why weren't the histology experiments carried out the brains of the animals either at the same age as those tested or following completion of the behavioural task, so that a more direct inference could be made between NVU/microglial changes and behavioural consequences? How do the authors know that the histological changes observed at P30 persisted at later ages? Moreover, how can the authors distinguish behavioural changes induced by alterations in vascular or microglial properties from those that may arise independently due to maternal HF-induced neuronal dysfunction?

We agree that studying adolescent behaviors would allow to have a more direct interactions of the phenotype we observe when assessed during the same period. We have previously observed in our maternal high-fat diet model mild functional behavioral alterations in social memory task and sensorimotor gating of the acoustic reflex during adolescence (Bordeleau et al., 2021). These alterations could have resulted or be accompanied by changes in motor or cognitive function. However, adolescence is really a short time window (P30-P45) in mouse life, which limited our capacity to conduct a thorough screening of the offspring behavioural outcomes. Our aim in the current study with the behavioural experiment was to determine the persisting, long-term phenotype observed in mHFD-exposed offspring. We have now better pointed out this in the method part of the manuscript as well as slightly reformulated our manuscript title in that direction.

“One last cohort [...] was used to evaluate functional behavioural outcomes in adulthood (P60-P85) to determine the long-term consequences of mHFD exposure on the offspring.”

It is indeed true that the changes in the vasculature might not have persisted throughout the offspring lifetime, and should be the subject of future studies. Adolescence is a critical stage of maturation that likely mediate, at least partially, both the adolescent phenotype we have previously reported (Bordeleau et al., 2021) as well as the long-term consequences of mHFD depicted here.

Neurovasculature and microglia are known to impact neuronal function. We found that mHFD affects neurovascular density as well as microglia and their interactions with the vasculature and synaptic elements (presynaptic terminals and postsynaptic dendritic spines which are likely to impact neuronal function). Then, these developmental changes should all be considered as whole, as together mediating the mHFD-induced functional behavioural changes.

4. A significant amount of work was done on microglia, but there was very little discussion of the current work in the context of previous studies looking at maternal HF and brain inflammation/microglia, despite the fact that the authors themselves have published on this topic (Bordeleau et al. *J Neuroinflammation*. 2020; 17: 264; Bilbo and Tsang *FASEB J* 2010 Jun;24(6):2104-15; Smith et al. *Brain, Behaviour and Immunity* 2020, 84:80-89). There was also no discussion of the observed regional variation in the effect of maternal HF on vessel density or microglial contact or discussion of why male and female offspring were equally affected by maternal HF when several other studies have suggested sex-specific susceptibility to prenatal HF exposure (for example, Edlow et al. *Am J Obstet Gynecol* 2016, 214, 623 e621-623 e610; Glendining and Jasoni *Int J Mol Sci* 2019;20(2); Vucetic et al. *Endocrinology* 2010 151(10), 4756-4764).

Given the length limitation of the short communication research article type of our previously submitted manuscript, we had decided to mainly focus the discussion on microglia-vasculature

interactions, which is an emerging concern in the mHFD field. We have now further discussed microglia as well as sex-specific susceptibility within the full article format.

“Therefore, it seems that cortical microglia of male offspring show a shift in their metabolism, motility as well as interactions with presynaptic axon terminals, while microglia from female offspring are less globally impacted but show modifications of cellular stress, thus highlighting sex-specific changes of microglia. Indeed, previous work by our team and others have also identified sex differences in the consequences of mHFD of the offspring throughout their life, and in other brain regions^{11,20,27–29}. Of note, regardless of the maternal diet, microglia presented sex difference in organelles involved in metabolism (i.e., mitochondrial alterations) and in phagocytosis (i.e., tertiary lysosomes, empty endosomes) (**Supplementary Figure 5, Supplementary Table 3**), further highlighting the sex difference of microglia and the importance to consider both sexes throughout developmental stages in future studies^{20,30–36}.”

5. More complete information is needed about the diet compositions (e.g. AFE from fat, protein, carbohydrates). Were the chow and HF diet otherwise matched for vitamins and other components that affect brain development (e.g. folic acid, iron etc)?

We have now added in supplementary information a table summarizing the diet compositions based on the available information from the supplier (Supplementary Table 5) and mention the similitude in caloric and protein intake between the diet groups.

“Throughout the diet protocol, dams consumed equivalent caloric and protein intake, while fat intake was increased to the detriment of carbohydrates, as previously reported (Bordeleau et al., 2020).”

Minor comments:

1. The manuscript should be reviewed for typos (e.g., Fig. 3, extended Fig. 5a) and formatting (e.g., Fig. 4).
We apologize for the overlook and have carefully reviewed the manuscript for typos and formatting issues. Thank you!

Reviewer #3 (Remarks to the Author):

In this manuscript, the authors investigated the effects of maternal HFD on neurovascular development, with a particular focus on microglial interactions with the blood vessels, as well as on a range of adult behaviours. This is an interesting study, integrating a range of novel methodologies with respect to the analysis of immunohistochemical data. I have, however, some major concerns, particularly regarding the statistical analyses.

Specific comments:

1. What was the rationale for choosing the adolescence as the first timepoint of observation? How does this time correspond to the developmental trajectory of the neurovascular system?
Thank you. The rationale for choosing adolescence as the first timepoint of observation was added to the Introduction section of the manuscript.
“Using an approach that combined state-of-the-art imaging and transcriptomic-based techniques, we evaluated the impact of mHFD within the mouse cerebral cortex and hippocampus during adolescence, which coincides with a critical stage of immune maturation and neurovascular network refinement (Lacoste et al., 2014; Brenhouse and Schwarz, 2016).”

2. Please provide more details about animal use. How many dams in total contributed to this study? Were there any effects of HFD on pregnancy and litter outcomes? Including body weight at birth, pup mortality and sex ratio?

We have now detailed the specific number of dams used and provided information on pregnancy rate and litter size. We also now specify the similitude in offspring body weight at weaning (P21) as well as the similar occurrence of developmental anomalies (unopened eyes, dwarfism, dental deformation) between offspring groups.

“A total of seven staggered cohorts of female mice were used for litter breeding (n = 86-114 females/diet). [...] CD and HFD dams had a rate of pregnancy of 38.4% and 51.8%, respectively, and both groups of dams showed similar pregnancy duration and litter size (number of total offspring, male and female pups). Of note, dams fed with HFD dams tended to cannibalize more often their litter without reaching statistical significance (37.3% vs 21.2%). [...] Weaned offspring had similar body weight and showed a similarly low prevalence of developmental anomalies, as previously observed (Bordeleau et al., 2020).”

3. The decision to pool the data when the effect of sex was not significant is questionable. It is equally important to show the direction of change is similar in males and females. If the study is sufficiently powered, then the effects of treatment should be similar in each sex alone, as when the data are pooled and the n is larger. Please present all data in the same manner, with sex as an independent variable.

We apologize for the confusion. All data analysis were conducted using 2-way ANOVAs for data meeting normality and homoscedasticity, or mixed-effects model for non-normally distributed data. Thus, the pooling of the data was purely for representation purposes, which we believed could have helped the reader easily visualize the main *Diet* main effect.

Considering the confusion that this created, we now present all data split by sex.

4. The results of the transcriptome analysis are unclear. Differences were detected in 15 transcripts, but out of how many in total? Can the authors include more detail about this array, including the analysis strategy (threshold fold change), and supplementary tables presenting changes in other genes that were detected in the array.

We apologize for the lack of clarity. The microarray analysis was conducted on the whole mouse genome, in which only 15 transcripts were found to be altered considering a threshold fold change of ± 2.0 . We now further detail the microarray method section along those lines:

“At P30, flash-frozen cortex of each offspring (n=4 animals/diet) were homogenized in Trizol (cat#15596-026, Ambion, Austin, TX, United States) and total RNA was extracted using the Trizol/chloroform method followed by an isopropanol precipitation. The RNA pellet was washed once in 75% ethanol, let dry, then reconstituted in Nuclease-free water (cat#AM9937, Ambion). After solubilization in RNase-free water, RNA was passed through high Pure PCR Cleanup micro kit following manufacturer’s guidelines (cat#04983912001, Roche, Basel, Switzerland) to eliminate potential contaminants.

Total RNA samples were sent to Genome-Québec for whole transcript expression analysis. Sample quantity was measured by NanoDrop Spectrophotometer ND-1000 (NanoDrop Technologies LLC., Wilmington, DE, United States) and integrity was evaluated using a 2100 Bioanalyzer (Agilent Technologies, Santa Clara, CA, United States). Sense-strand cDNA was synthesized from 100ng of total RNA. According to the manufacturer’s instructions (Thermo Fisher Scientific), ssDNA were produced by fragmentation and labeling using GeneChip® whole transcriptome terminal labeling

kit. 3.5 µg of cDNA was hybridized on Clariom™ S mouse array (Thermo Fisher Scientific) for 17h at 60 rpm. Clariom™ S mouse arrays were washed with GeneChip® Fluidics Station 450 (Thermo Fisher Scientific) using GeneChip Hybridization Wash and Stain kit. Finally, microarrays were scanned on GeneChip® scanner 3000 (Thermo Fisher Scientific) and signal was analyzed for the whole mouse genome using the Transcript Analysis Console (TAC) 4.0 software, generating differentially expressed gene lists, where threshold fold change was ± 2.0 (Applied Biosystems, Thermo Fisher Scientific).”

5. What was the rationale to conduct all the different behavioural tests in this study? What is the role of neurovascular function in underlying the development of these behaviours? Were all behaviours conducted in the same set of animals? What was the order of the tests? The authors mention neurodevelopmental disorders as the precedence for behavioural testing. However, the tests were conducted in adulthood and not during early development, when neurodevelopmental disorders are likely to emerge.

Our aim with the behavioural experiment was to determine the persisting phenotype observed in mHFD-exposed offspring at adulthood. Although neurodevelopmental disorder phenotypes emerge and are diagnosed early during post-natal development, their symptoms persist during the patient’s lifespan. Moreover, studying early post-natal stages or adolescence limit the number and type of behaviours that can be assessed, thus limiting thorough screening of functional outcomes on the offspring. More details of the rationale behind the different behavioural tests performed and the protocols used can now be found in the Methods section:

“One last cohort of offspring (composed of four staggered breeding cohorts, n = 6-10 litters/diet/sex, N = 8-12 animals) was used to evaluate functional behavioural outcomes in adulthood (P60-P85) to determine the long-term consequences of mHFD exposure on the offspring.”

“At adulthood (P60-P85), offspring underwent behavioural assessment, from the less to more stressful test, and with 2 days of rest in between. A thorough characterization of the functional outcomes including a vast range of behaviours known to be altered in neurodevelopmental disorders (e.g., motor, social, cognitive, mood, sensorial) in the animal life course, including adulthood (Tanimizu et al., 2017; Sun et al., 2020; Shin and Liberzon, 2010; Tapias-Espinosa et al., 2019; Miller et al., 2010), was performed. General motricity was assessed by open field test. Social preference and social novelty preference were measured by three-chambers social interactions. Novel object recognition was used to evaluate spatial memory. Marble-burying test was used to assess repetitive and stereotypic behaviours. Anxiety under normal conditions was gauged using the open field and elevated plus maze tests. Sensorimotor gating of the acoustic reflex was measured by the prepulse inhibition test.”

6. Overall, out of the 6 tests, with each test providing diverse information about a range of behaviours, differences were detected only in the marble burying task. These data suggest there were overall minimal long-term effects of maternal HFD, despite changes in neurovascular parameters in adolescence. I therefore suggest the authors to rewrite their conclusions emphasising these encouraging findings, rather than suggesting the effects of maternal HFD were detrimental.

Thank you for your suggestion. We have now reworded the conclusion of our work towards this, indeed, encouraging finding.

“Overall, our work reveals the profound impact that excess of fats, through mHFD exposure, exerts on the development of the offspring’s neurovascular system, leading to changes in vascular structure and microglial-vessel coverage, metabolism- and immune-regulating gene expression. While we observed stereotypic and repetitive behaviours in adulthood, most behavioural parameters remained

normal suggesting the encouraging possibility that mHFD-induced vascular and microglial alterations may be part of a beneficial adaptation that results in a relatively minimal impact on the long-term phenotype. That being said, similar mHFD-driven cerebrovascular modifications of the NVU in humans could render, at least in part, the offspring prone to developing a wide variety of neurodevelopmental and neuropsychiatric disorders, which warrants further investigation of the combined impact of environmental and genetic factors. Hence, appraising the influence of mHFD on the cerebrovasculature stresses the importance of implementing precautionary measures to notably reduce the “priming” effect that mHFD may have on the progeny, while studying the mechanisms involved and identifying therapeutic targets for children and adolescents at risk.”

7. The authors incorporate interesting microglial analyses in this study. Could the authors indicate whether in addition to changes in microglia-blood vessel proximity were there also changes in microglial numbers? No changes in microglial density suggest there were no difference in their activation state, but could the authors elaborate about this in the manuscript.

After revision, we complemented previous microglial analyses with a thorough characterization of cortical microglial ultrastructural interactions and organelles, which allowed us to further discuss microglial state upon mHFD. Moreover, from our previous work using the same model and time point in other brain regions, namely the corpus callosum and the dorsal hippocampus, we have shown that even without microglial density changes, microglial state was changed, as highlighted by alterations in their morphology, the expression of genes enriched in microglia as well as their ultrastructural interaction with the microenvironment (Bordeleau et al., 2020, 2021).

8. Language editing is recommended.

We apologize for the overlook and have carefully reviewed the manuscript for typos and formatting issues. Thank you!

REVIEWERS' COMMENTS:

Reviewer #2 (Remarks to the Author):

The manuscript by Bordeleau has been significantly improved by the additional experiments and the authors should be commended for the amount of data they have collected. Most of the previous comments have been addressed.

However, there is still some concern that the authors are drawing inferences about biological pathways altered by mHFD based on the transcriptomics data which is still largely unvalidated. For example, on p. 10, the authors state that "out of all the transcriptomic changes, Igtp expression was the most strongly increased after exposure to mHFD". However, the fold change for Igtp between mHFD and CD was only 2.01, compared to -5.58 and 16.48 for Hba and Erdr1, respectively. Presumably, the authors are referring to p values, and if so, this should be clarified. However, expression of Erdr1 in microglia wasn't altered between CD and mHFD mice, indicating that, as originally queried, it is difficult to draw conclusions about the role that these genes/proteins/pathways have in underlying the observed vascular and/or behavioural changes. Therefore, some of the conclusions relating to these data should be moderated to reflect this uncertainty.

Related to this, the last sentence of Introduction starting with "Together, these changes contributed to the development of stereotypic" is a bit of an overstatement, given that neuronal changes have been reported in the offspring brain following maternal HF and not examined here. This should be rephrased to be more speculative.

More minor points relate to typos/sentence structure that still remain (e.g. pg. 6 "evenly distribution", p.11 "these cues were balanced endothelial cell"), particularly in the Methods section (e.g. p.13 "Animals were live dislocated... and their brain. Brain hemisphere were split", "For transcriptomic, offspring was anesthetized...")

Reviewer #3 (Remarks to the Author):

The authors have made substantial changes to answer the reviewers' comments, including the addition of microglial ultrastructural analyses and FISH-immunofluorescence, and the manuscript has been substantially improved.

Few minor comments:

Please clarify in the statistical analyses description in the methods for which analyses have the authors considered individual animals as opposed to individual microglia cells.

In supplementary tables, what do the authors mean by the following: "For each data, Mean \pm SEM are giving each group of offspring"? Is it meant to be 'represent each treatment group'?

Supplementary table 5 – not sure why is there 'g/kg' on line 3, next to the 'units' column (directly underneath the CD and HFD headings)?

Figure 3 – e and f in the figure legend are reported in the opposite order to their representation in the figure.

Page 13, offspring were, not 'was'. Also, please amend transcriptomic to 'transcriptomics' or 'transcriptomic analyses' in the same sentence and in the heading on page 19.

REVIEWERS' COMMENTS:

Reviewer #2 (Remarks to the Author):

The manuscript by Bordeleau has been significantly improved by the additional experiments and the authors should be commended for the amount of data they have collected. Most of the previous comments have been addressed.

Thank you for acknowledging our desire and effort to address your previous comments as much as possible.

- However, there is still some concern that the authors are drawing inferences about biological pathways altered by mHFD based on the transcriptomics data which is still largely unvalidated. For example, on p. 10, the authors state that “out of all the transcriptomic changes, Igtp expression was the most strongly increased after exposure to mHFD”. However, the fold change for Igtp between mHFD and CD was only 2.01, compared to -5.58 and 16.48 for Hba and Erdr1, respectively. Presumably, the authors are referring to p values, and if so, this should be clarified. However, expression of Erdr1 in microglia wasn't altered between CD and mHFD mice, indicating that, as originally queried, it is difficult to draw conclusions about the role that these genes/proteins/pathways have in underlying the observed vascular and/or behavioural changes. Therefore, some of the conclusions relating to these data should be moderated to reflect this uncertainty.

Thank you for pointing the lack of clarity, we have modified *as per* suggested.

“Out of all the transcriptomic changes, Igtp expression was the most strongly robustly increased after exposure to mHFD, as indicated by its lower *P* value. Moreover, its expression was confirmed to be expressed by a majority of Iba1+ cortical microglia (CD: 80.5% of microglia vs mHFD: 83.8% of microglia), as revealed by fluorescence *in situ* hybridization combined with immunofluorescence labelling (FISH-immunofluorescence; Figure 4d). [...] However, it remains difficult to discriminate or conclude how these different pathways or genes are involved in underlying the observed vascular changes, which would warrant future studies to specifically look at these different targets.”

- Related to this, the last sentence of Introduction starting with “Together, these changes contributed to the development of stereotypic” is a bit of an overstatement, given that neuronal changes have been reported in the offspring brain following maternal HF and not examined here. This should be rephrased to be more speculative.

We have now reformulated to be more speculative: “Together, these changes may have contributed in part to the development of stereotypic and repetitive behaviours assessed by marbles burying at adulthood, suggesting that mHFD-mediated neurovascular alterations might in part be involved in the expression of behaviours reminiscent of neurodevelopmental disorders¹⁴.”

- More minor points relate to typos/sentence structure that still remain (e.g. pg. 6 “evenly distribution”, p.11 “these cues were balanced endothelial cell”), particularly in the Methods section (e.g. p.13 “Animals were live dislocated... and their brain. Brain hemisphere were split”, “For transcriptomic, offspring was anesthetized...”)

Thank you for mentioning, we have corrected these minor errors.

Reviewer #3 (Remarks to the Author):

The authors have made substantial changes to answer the reviewers' comments, including the addition of microglial ultrastructural analyses and FISH-immunofluorescence, and the manuscript has been substantially improved.

Thank you for the positive assessment of this revised version as well as your helpful comments that helped to improve our work.

Few minor comments:

- Please clarify in the statistical analyses description in the methods for which analyses have the authors considered individual animals as opposed to individual microglia cells.
We have now incorporated this information in the statistical analyses section:
“Sample size was defined by the number of individual animals for all analyses except for ultrastructure analyses in which sample size represented the number of individual microglial cells or capillaries.”
- In supplementary tables, what do the authors mean by the following: “For each data, Mean \pm SEM are giving each group of offspring”? Is it meant to be ‘represent each treatment group’?
This has been changed to: “Data are presented as Mean \pm SEM for each treatment group.”
- Supplementary table 5 – not sure why is there ‘g/kg’ on line 3, next to the ‘units’ column (directly underneath the CD and HFD headings)?
This is now clarified in the title of the Supplementary Table 5:
“The different components of the diet are presented as either a percentage of food or a quantity (g, mg or μ g) per kilogram of food.”
- Figure 3 – e and f in the figure legend are reported in the opposite order to their representation in the figure.
Thank you for pointing this out. It is now appropriately reported.
- Page 13, offspring were, not ‘was’. Also, please amend transcriptomic to ‘transcriptomics’ or ‘transcriptomic analyses’ in the same sentence and in the heading on page 19.
Thank you for pointing out the overlook. These have now been modified accordingly.